# Pancreatic circulating tumor cell profiling identifies LIN28B as a metastasis driver and drug target

Joseph W. Franses[1,6], Julia Philipp[1,6], Pavlos Missios[2], Irun Bhan [3], Ann Liu [1], Chittampalli Yashaswini[1], Eric Tai[1], Huili Zhu[1], Matteo Ligorio[1], Benjamin Nicholson[1], Elizabeth M. Tassoni[1], Niyati Desai[1], Anupriya S. Kulkarni[1], Annamaria Szabolcs [1], Theodore S. Hong[1], Andrew S. Liss [4], Carlos Fernandez-del Castillo[4], David P. Ryan[1], Shyamala Maheswaran[1], Daniel A. Haber[1,5], George Q. Daley[2] & David T. Ting [1]✉

Pancreatic ductal adenocarcinoma (PDAC) lethality is due to metastatic dissemination. Characterization of rare, heterogeneous circulating tumor cells (CTCs) can provide insight into metastasis and guide development of novel therapies. Using the CTC-iChip to purify CTCs from PDAC patients for RNA-seq characterization, we identify three major correlated gene sets, with stemness genes *LIN28B/KLF4*, *WNT5A*, and *LGALS3* enriched in each correlated gene set; only *LIN28B* CTC expression was prognostic. CRISPR knockout of *LIN28B*—an oncofetal RNA-binding protein exerting diverse effects via negative regulation of let-7 miRNAs and other RNA targets—in cell and animal models confers a less aggressive/metastatic phenotype. This correlates with de-repression of let-7 miRNAs and is mimicked by silencing of downstream let-7 target *HMGA2* or chemical inhibition of LIN28B/let-7 binding. Molecular characterization of CTCs provides a unique opportunity to correlated gene set metastatic profiles, identify drivers of dissemination, and develop therapies targeting the "seeds" of metastasis.

[1] Massachusetts General Hospital Cancer Center, Harvard Medical School, Boston, MA 02114, USA. [2] Children's Hospital Boston, Harvard Medical School, Boston, MA 02115, USA. [3] Massachusetts General Hospital Division of Gastroenterology, Harvard Medical School, Boston, MA 02114, USA. [4] Massachusetts General Hospital Department of Surgery, Harvard Medical School, Boston, MA 02114, USA. [5] Howard Hughes Medical Institute, Chevy Chase, MD 20615, USA. [7] These authors contributed equally: Joseph W. Franses, Julia Philipp. ✉email: DTING1@mgh.harvard.edu

Pancreatic ductal adenocarcinoma (PDAC) remains one of the most lethal solid tumors, with a long-term survival rate of ~8%[1]. Only ~20% of patients are eligible for upfront surgical resection, which is rarely curative and is frequently associated with recurrence of highly aggressive disease within months[2]. Distant spread can occur early, with the majority of patients with even a single involved lymph node developing distant metastases[3]. A major obstacle in the treatment of PDAC is the incomplete understanding of the mechanisms of PDAC invasion and metastasis and the paucity of actionable drug targets[4]. Although outcomes have been improved with the use of more effective preoperative[5] or postoperative[6] therapy, identifying the possible drivers of early recurrence and developing therapeutics to inhibit metastatic dissemination could fundamentally change the course of this disease.

Circulating tumor cells (CTCs) necessarily contain the precursors of distant metastatic foci[7], and the analysis of these rare cells can provide unique mechanistic insights into the metastatic cascade and inform therapeutic target development. We have extensive experience utilizing the CTC-iChip, which has been refined into a clinical-grade platform for surface marker-agnostic CTC purification from the blood of patients with solid malignancies[8]. Unlike more commonly used commercially-available CTC purification technologies like CellSearch[9] that rely on positive selection of CTCs via a pre-specified capture antigen such as EpCAM, the CTC-iChip relies on depletion of all other known blood cell types and hence purifies CTCs without biases for particular antibody-mediated capture antigens. Our group has utilized multiple versions of this device to identify multiple PDAC CTC targets by RNA-sequencing profiling of purified CTCs. We have previously identified Wnt signaling[10] and aberrant extracellular matrix (ECM) expression[11] as orthogonal drivers of the PDAC metastatic phenotype utilizing primarily CTCs collected from genetically-engineered murine cancer model systems. Although animal model systems can provide deep insights into disease biology, the development of mouse PDAC in the setting of whole-pancreas genetic lesions during embryogenesis creates advanced tumors that are different from early human PDAC tumors that emerge stochastically at later ages.

It is understandable that much of the focus on CTC biology has been in the advanced or metastatic settings, a setting in which there should undoubtedly exist many metastatic precursors present within the blood. However, the study of early CTC biology provides an opportunity to understand similarities and differences in the mode of metastasis in early versus late disease. The presence of early CTCs was first demonstrated in genetically-engineered mouse models of preneoplastic pancreatic intraepithelial neoplasia (PanIN)[12]. Following these provocative mouse studies, we[13] and others[14] identified cells released from precancerous human pancreatic intraductal papillary mucinous neoplasm (IPMN) tumors in quantities similar to those released by localized early PDACs. Consistent with these findings, CTCs have been isolated from patients with early-stage prostate cancers[15], most of whom never proceeded to develop overt metastatic disease. Altogether, these data imply that entry into circulation is not a rate-limiting step for metastasis formation and suggest an intervention in early CTC biology offers the potential to increase curability of localized PDAC patients.

In the current study, we utilize RNA-sequencing of CTCs purified from the blood of patients with treatment-naïve localized PDAC (locPDAC) and metastatic PDAC (metPDAC) to gain insights into the drivers of CTC biology. Gene expression profiling of the PDAC CTCs identifies three major correlated gene sets expressing different sets of stemness genes. In the largest PDAC CTC subgroup, we noted robust expression of LIN28B, an oncofetal RNA-binding protein that exerts diverse effects primarily via negative regulation of let-7 microRNA (miRNA) maturation[16] and whose expression correlates with poor outcome in many tumor types including PDAC[17]; our data also show a negative impact of LIN28B PDAC CTC expression and overall survival. Using cell and animal models we define a mechanistic contribution of LIN28B in metastatic fitness and we provide proof of concept of utilizing a LIN28B small molecule inhibitor to disrupt CTC metastatic efficiency. Taken together, our work demonstrates the potential of translating CTC expression signatures to select specific patients that may benefit from novel therapeutics targeting the specific molecular pathways enriched in CTCs.

## Results

**Pancreatic CTC RNA-seq expression profiles are heterogeneous.** CTCs are enriched for putative metastatic precursors and their characterization provides a window into the biology of these rare cells. The CTC-iChip utilizes a combination of deterministic lateral displacement of nucleated cells away from smaller platelets and erythrocytes and efficient immunomagnetic depletion of pre-labeled leukocytes to purify rare CTCs from whole blood by a factor of $10^4$ or higher[8]. This technique does not rely on antigen-mediated capture of CTCs—which may express heterogeneous levels of any given combination of cell surface markers—but rather by exceptionally efficient removal of known hematopoietic cells. We have previously utilized this technology to enumerate CTCs released from both localized PDAC and premalignant IPMN lesions[13]. In this work, we utilized CTCs purified from patients with never-treated localized PDAC ($n = 17$; Table 1) and metastatic PDAC ($n = 18$; Table 2); purified healthy donor (HD, $n = 21$) volunteer blood was utilized as a control (Fig. 1a; Tables 1–2, Supplementary Table 1). After purification, CTC enriched cell suspensions were flash-frozen and stored for batched RNA isolation and bulk RNA-sequencing.

Commonly utilized epithelial markers (keratins KRT7/KRT8/KRT18/KRT19/KRT23[11,18]) were variably detected by our gene expression analysis (Fig. 1b, Supplementary Fig. 1a), with some background present in HD purified cells consistent with the presence of rare circulating epithelial cells that are not PDAC-specific. Individual keratin (KRT) genes, in general, were co-expressed with each other, but there was notable variability in detection of KRT genes in a given sample. This could reflect phenotypic plasticity of CTCs on a spectrum of epithelial-mesenchymal transition (EMT), a phenomenon that we have previously identified with single CTC RNA-seq analysis[11]. Mucin (MUC) genes are well known to be expressed in diseased pancreatic tissue[19–22], but not in normal pancreas tissue (Supplementary Fig. 1b; GTEX portal) and expression of the MUC3A, MUC4, MUC16, and MUC17 genes were significantly upregulated in locPDAC and metPDAC compared to HD (DESeq2 FDR < 0.10; Fig. 1c). To account for variation of KRT and MUC genes due to biological changes (EMT) or technical limits of detection, we used the sum of the KRT or MUC genes as metasignatures of CTC detection, but only the MUC metasignature was significantly higher for locPDAC and metPDAC relative to HD controls ($p < 0.05$; Supplementary Fig. 1a). Notably, KRT and MUC gene expression were significantly correlated in CTCs obtained from patients with PDAC (Supplementary Fig. 1c), but not in cells purified from HD controls, indicating a correlation of circulating epithelial signal with pancreas-specific markers in patients with pancreatic lesions. The mucin metasignatures did not predict survival in our cohort of patients (Supplementary Fig. 1d), suggesting that the presence of CTCs is not directly linked with clinical outcome. Analysis of Z-score normalized TCGA data also demonstrated variable expression of KRT and

**Table 1 Localized PDAC patient characteristics at the time of blood draw for CTC purification.**

| ID | Age (yrs) | Sex | CA19-9 (U/mL) | CEA (ng/mL) | T stage | N stage | M stage | Stage |
|---|---|---|---|---|---|---|---|---|
| locPDAC01 | 82 | M | 224 | 3.6 | 2 | 2 | 0 | III |
| locPDAC02 | 49 | F | 566 | 1.1 | 2 | 0 | 0 | I |
| locPDAC03 | 70 | M | 40 | 4.7 | 1 | 0 | 0 | I |
| locPDAC04 | 72 | M | 82 | 5 | 1 | 2 | 0 | II |
| locPDAC05 | 65 | M | 1943 | 4.8 | 2 | 0 | 0 | I |
| locPDAC06 | 74 | F | 289 | 2.5 | 3 | 0 | 0 | II |
| locPDAC07 | 59 | F | 418 | 2.9 | 2 | 0 | 0 | I |
| locPDAC08 | 61 | M | 179 | 2 | 1 | 1 | 0 | II |
| locPDAC09 | 60 | M | 132 | 3.2 | 2 | 0 | 0 | I |
| locPDAC10 | 89 | F | 11.7 | 24 | 3 | 0 | 0 | II |
| locPDAC11 | 50 | F | 1040 | 8.3 | 2 | 0 | 0 | I |
| locPDAC12 | 66 | F | 2 | 1.5 | 1 | 0 | 0 | I |
| locPDAC13 | 83 | F | 19 | 1.5 | 1 | 2 | 0 | II |
| locPDAC14 | 69 | M | 63 | 4.9 | 2 | 1 | 0 | II |
| locPDAC15 | 69 | F | 369 | 5.4 | 3 | 2 | 0 | III |
| locPDAC16 | 76 | M | 15 | 3.2 | 2 | 0 | 0 | I |
| locPDAC17 | 75 | F | 70 | 2.1 | 2 | 0 | 0 | I |

MUC genes in histologically verified PDAC, which supports the intrinsic heterogeneity of PDAC tumor cells (Supplementary Fig. 1e). Gene ontology analysis of genes that were differentially upregulated in CTCs isolated from patients with locPDAC and metPDAC compared with cells purified from HD controls (Supplementary Table 2) revealed several expected signatures including KRAS, TGF-beta, and proliferative signatures (Fig. 1d). Altogether, these findings confirm that entry into circulation can be an early event, with shared transcriptional programs that are not rate limiting steps for cancer cell distant spread.

**Human PDAC CTC correlated gene sets are enriched for stemness genes**. To determine whether there were PDAC CTC correlated gene sets that can be observed with our technology, we performed gene correlation analysis across all genes that were enriched in blood from either locPDAC or metPDAC patients compared with HDs; we identified three major subgroups of correlated genes within PDAC CTCs (Fig. 2a). Since many gene programs that drive metastasis can be associated with stemness phenotypes[23] we focused on stemness-related genes that were enriched in each major subgroup. We evaluated a panel of genes that have been reported as PDAC stem cell markers—including WNT genes, Hedgehog (SHH), Notch, c-Met, ALDH1, LGR5, CD133, CD24, CD44, CXCR4, EPCAM, and LGALS3[24]—and known cellular reprogramming genes (MYC, OCT4, SOX2, NANOG, KLF4, LIN28A/B)[25]. This analysis identified an enrichment in stemness genes in correlated gene set 1 of several co-expressed PDAC CTC genes, including LIN28B, KLF4, CD24, multiple ALDH genes, and multiple WNT ligands; WNT5A appeared in correlated gene set 2 and LGALS3 in correlated gene set 3 (Supplementary Table 3).

When clustered by sample using the top 20 genes correlated with LIN28B, KLF4, WNT5A, and LGALS3, there were still three major subgroups of samples: LGALS3-high, WNT5A-high, and LIN28B-high, with high KLF4 expression being shared between the WNT5A and LIN28B groups (Fig. 2b). In order to validate CTC expression of our newly defined correlated gene sets, we performed similar CTC RNA-seq analysis on a separate cohort of 25 patients collected in an unrelated prospective therapeutic clinical trial (Supplementary Fig. 2) and found that similar sets of genes correlated with our selected stemness factors. Each of the above stem cell markers can be found in both locPDAC and metPDAC, indicating that they are not exclusive to early versus late disease presentation. Our previous work identified LGALS3,

**Table 2 Metastatic PDAC patient characteristics at the time of blood draw for CTC purification.**

| ID | Age | Sex | CA19-9 (U/mL) | CEA (ng/mL) | Treatment status |
|---|---|---|---|---|---|
| metPDAC01 | 57 | M | 7 | 1.1 | Responding |
| metPDAC02 | 49 | F | 117 | 4.8 | Responding |
| metPDAC03 | 53 | F | 706 | 5.5 | Progressing |
| metPDAC04 | 58 | F | 1307 | 54 | Responding |
| metPDAC05 | 66 | F | 7268 | 2.6 | Progressing |
| metPDAC06 | 55 | M | 4 | – | Progressing |
| metPDAC07 | 60 | M | 354 | 2.9 | Responding |
| metPDAC08 | 66 | M | 355 | 3.4 | Responding |
| metPDAC09 | 70 | F | 18 | 1.5 | Responding |
| metPDAC10 | 75 | F | <1 | 11 | Responding |
| metPDAC11 | 49 | M | 32 | 6.4 | Responding |
| metPDAC12 | 66 | M | 174 | 19 | Responding |
| metPDAC13 | 71 | F | 39 | 5 | Progressing |
| metPDAC14 | 70 | F | 54 | 9.9 | Progressing |
| metPDAC15 | 71 | F | 49 | 1.6 | Responding |
| metPDAC16 | 79 | F | 19 | – | Responding |
| metPDAC17 | 67 | M | 593 | 4.2 | Naive |
| metPDAC18 | 68 | F | 39 | 2 | Naive |

KLF4, and non-canonical WNTs (i.e., WNT5A), as genes enriched in CTCs from overtly metastatic pancreatic, breast, and prostate cancer mouse models and patients[10,11,26]. LIN28B has been noted as a poor prognostic marker in a cohort of resected PDAC[17]. Analysis of TCGA data from resected PDAC revealed correlated expression of LIN28B pathway genes HMGA2 and IGF2BP1/2/3 (Supplementary Fig. 3a), supporting the notion of coordinate expression of these genes driven by LIN28B. Further, high expression of HMGA2 in this dataset correlated with poor clinical outcomes (Supplementary Fig. 3b). However, the correlated gene families that we identified in our CTC data were not observed in the TCGA dataset (Supplementary Fig. 3c), which likely stems from differences between primary tumor and CTC cellular heterogeneity as we had demonstrated previously with single cell RNA-seq[11].

Notably, other classic pluripotent reprogramming factors (MYC, OCT4, NANOG, and SOX2) were not enriched in our PDAC CTCs (Supplementary Fig. 4a). Additionally, LIN28B was not previously seen in our CTC analysis of PDAC metastatic

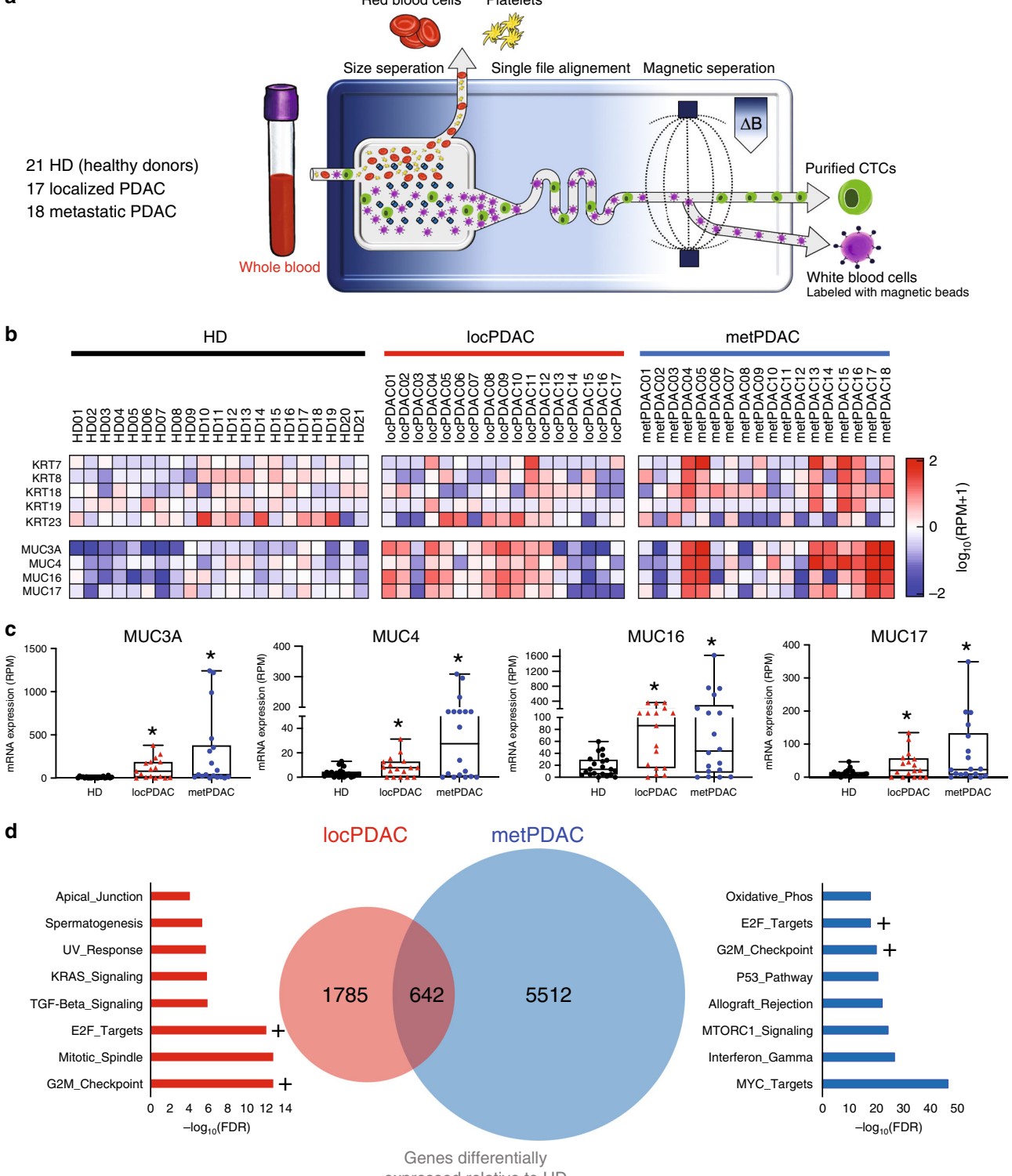

**Fig. 1 Transcriptional profiling of CTCs purified from the blood of patients with pancreatic cancer reveals substantial heterogeneity. a** Schematic of circulating tumor cell purification, via removal of erythrocytes and platelets followed by efficient magnetophoretic depletion of labeled leukocytes, utilizing the CTC-iChip. **b** Heatmap of $\log_{10}(RPM+1)$ mRNA expression of keratin and mucin genes in CTCs purified from the blood of patients with localized PDAC (locPDAC, $n = 17$), and metastatic PDAC (metPDAC, $n = 18$) compared with healthy donor (HD, $n = 21$) controls. **c** Box plots showing expression of 4 mucin genes detected in cells released from localized PDAC and metastatic PDAC. *FDR < 0.10 relative to HDs of DESeq2-normalized data; $n = 21$ HD, 17 locPDAC, 18 metPDAC; median and 25–75% IQR shown by box plots. **d** Venn diagram of genes upregulated (17 locPDAC and 18 metPDAC compared with specimens from 21 HD controls; *FDR < 0.10) in localized PDAC and metastatic PDAC CTCs; "Hallmark" gene set ontology terms for each group are shown. + Denotes common ontology terms shared between both groups.

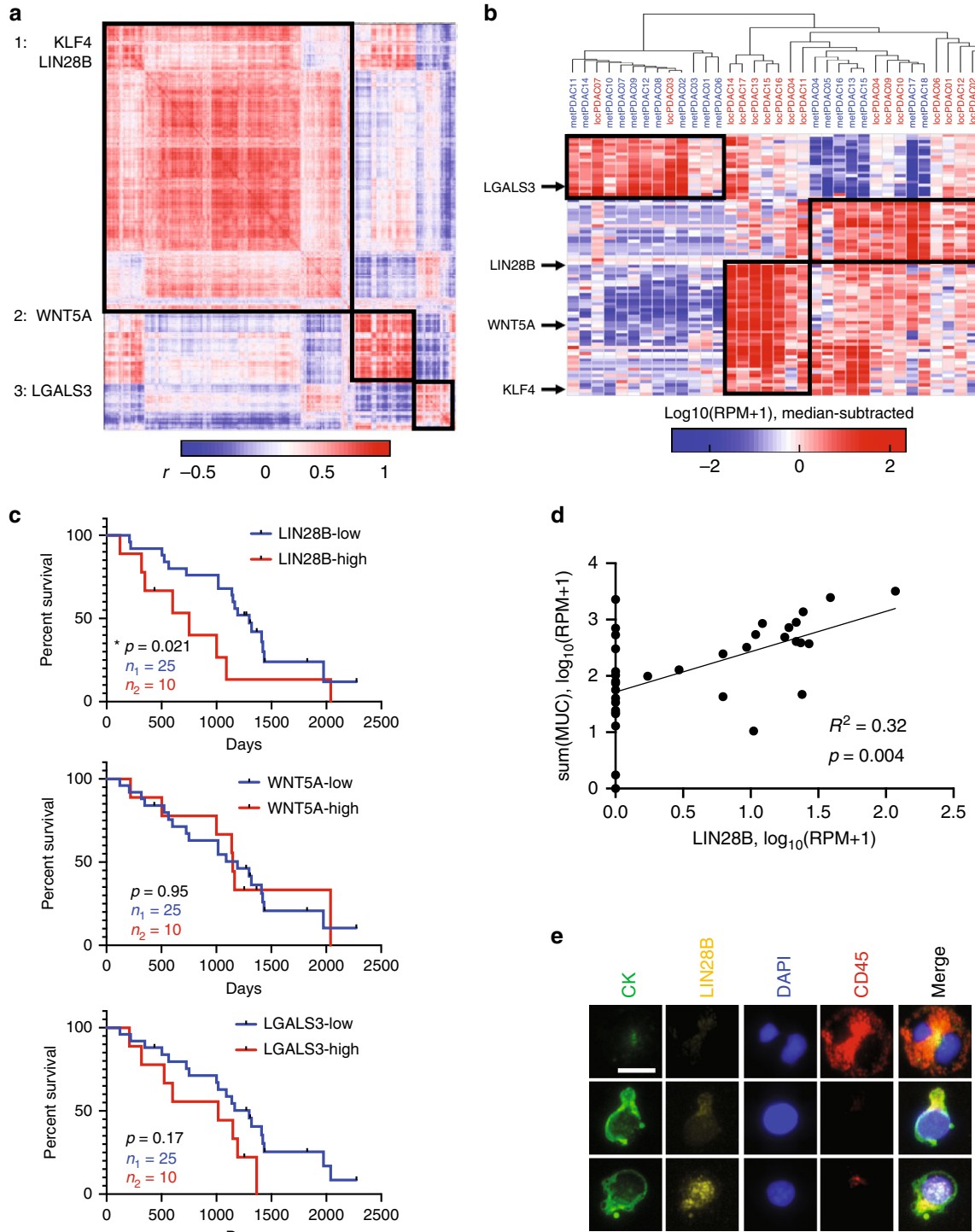

**Fig. 2 Transcriptional profiling of CTCs released from pancreatic cancer reveals three major correlated gene sets and the largest is enriched in LIN28B and KLF4. a** Hierarchical clustered correlation matrix of all genes detected in either locPDAC or metPDAC CTCs. **b** Hierarchical clustering of PDAC by sample utilizing the major genes driving each correlation subgroup from (**a**). **c** Kaplan–Meier curves showing overall survival for PDAC patients stratified based on CTC RNA expression of LIN28B, WNT5A, and LGALS3; "high" denotes highest quartile and "low" bottom 3 quartiles. *$p = 0.021$ by Gehan–Breslow–Wilcoxon test, $n = 25$ in the "low" group and 10 in the "high" group. **d** Log-log scatter plot and linear correlation of the sum of MUC3A/4/16/17 and LIN28B mRNA (RPM) in CTCs purified from patients with locPDAC and metPDAC. *$p = 0.004$ by least squares linear regression, $n = 35$. **e** Immunofluorescent images of a leukocyte (top row), LIN28B-low CTC (middle row), and LIN28B-high CTC (bottom row) from a separate cohort of patients with metastatic PDAC. Scale bar = 10 mm. This experiment was not repeated but showed consistent results in the specimens used.

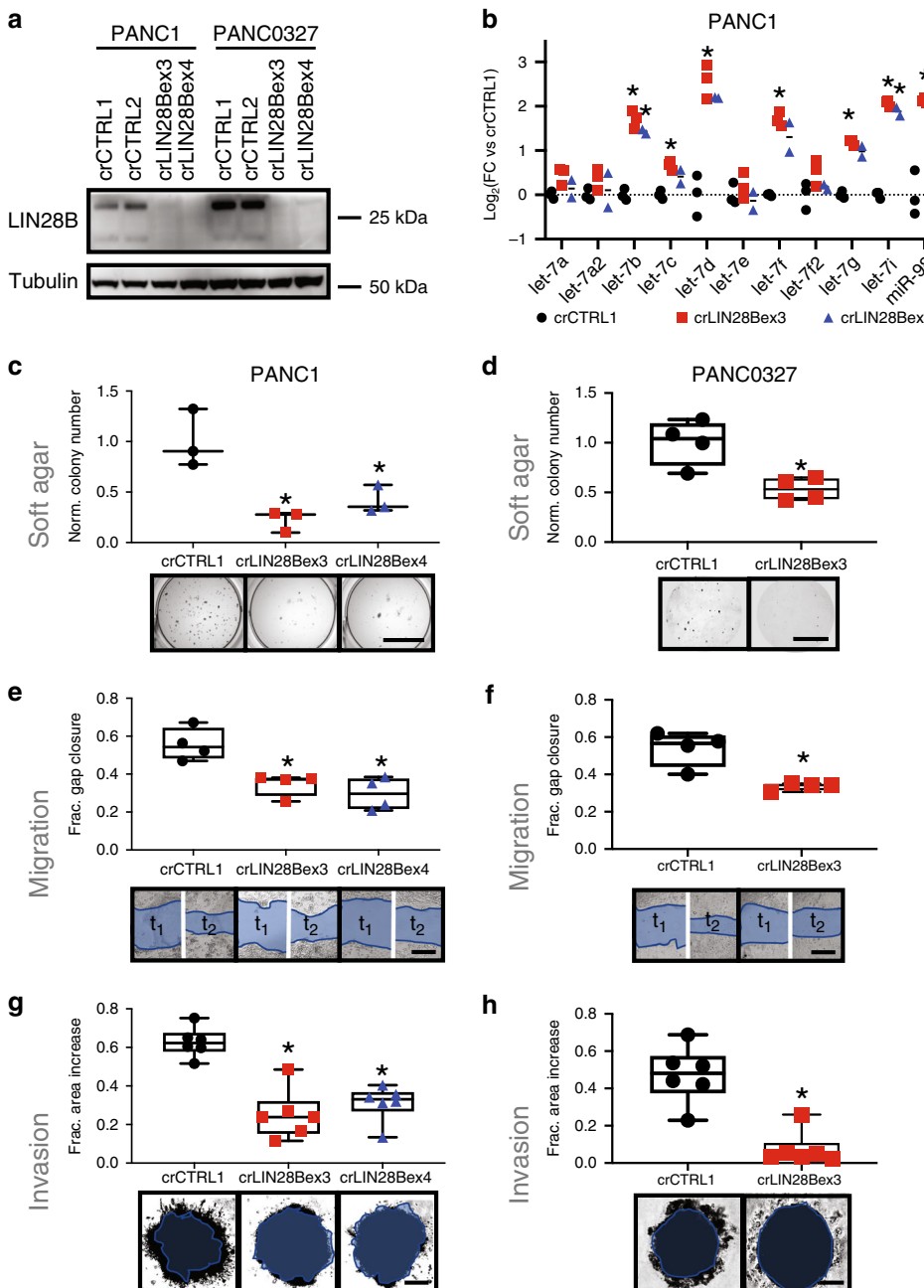

**Fig. 3 CRISPR knockout of LIN28B in LIN28B-high PDAC cells causes a less aggressive phenotype. a** Western blot of PANC1 and PANC0327 cell lines after stable lentiviral transduction with either non-targeting gRNA (crCTRL1, crCTRL2) or gRNAs targeting LIN28B exon 3 (crLIN28Bex3.1) or exon 4 (crLIN28Bex4). These results were confirmed with at least 2 replicate assays. **b** qPCR for mature let-7 species in PANC1 LIN28B-knockout cells versus control cells. *$p < 0.05$, 2-sided $t$-test with Holm-Sidlak multiple testing adjustment, $n = 3$ per group. **c, d** Quantification and representative images taken from soft agar colony formation assays for PANC1 (**c**) or PANC0327 (**d**) cells after generation of stable LIN28B CRISPR knockout. *$p < 0.05$, 2-sided $t$-test using Bonferroni adjustment, $n = 3$–4 per group, median and 25–75% IQR shown by box plots. Scale bar = 10 mm. **e, f** Quantification and representative images of two-dimensional scratch migration assays for PANC1 (**e**) and PANC0327 cells (**f**) after generation of stable LIN28B CRISPR knockout. *$p < 0.05$, 2-sided $t$-test using Bonferroni adjustment, $n = 4$ per group, median and 25–75% IQR shown by box plots. Scale bar Is 500 mm. **g, h** Quantification and representative images of three-dimensional invasion through extracellular matrix gels for PANC1 (**g**) and PANC0327 cells (**h**) after generation of stable LIN28B CRISPR knockout. *$p < 0.01$, 2-sided $t$-test using Bonferroni adjustment, $n = 6$ per group, median and 25–75% IQR shown by box plots. Scale bar = 300 mm (**g**) and 200 mm (**h**).

mouse models and PDAC patients[11]. Furthermore, high *LIN28B* expression—defined as the highest quartile in our dataset, which was greater than or equal to 16.5 reads per million (RPM)—in PDAC CTCs was associated with shorter patient survival, whereas PDAC CTC expression of *WNT5A* and *LGALS3* were not found to be adverse prognostic factors (Fig. 2c;

Supplementary Table 4). These features pointed to *LIN28B* as a newly identified PDAC CTC driver gene that merited further investigation.

To confirm that *LIN28B* expression was specific to purified PDAC CTCs, we first observed significant correlation with mucin gene expression (Fig. 2d, Supplementary Fig. 4b), and this was not

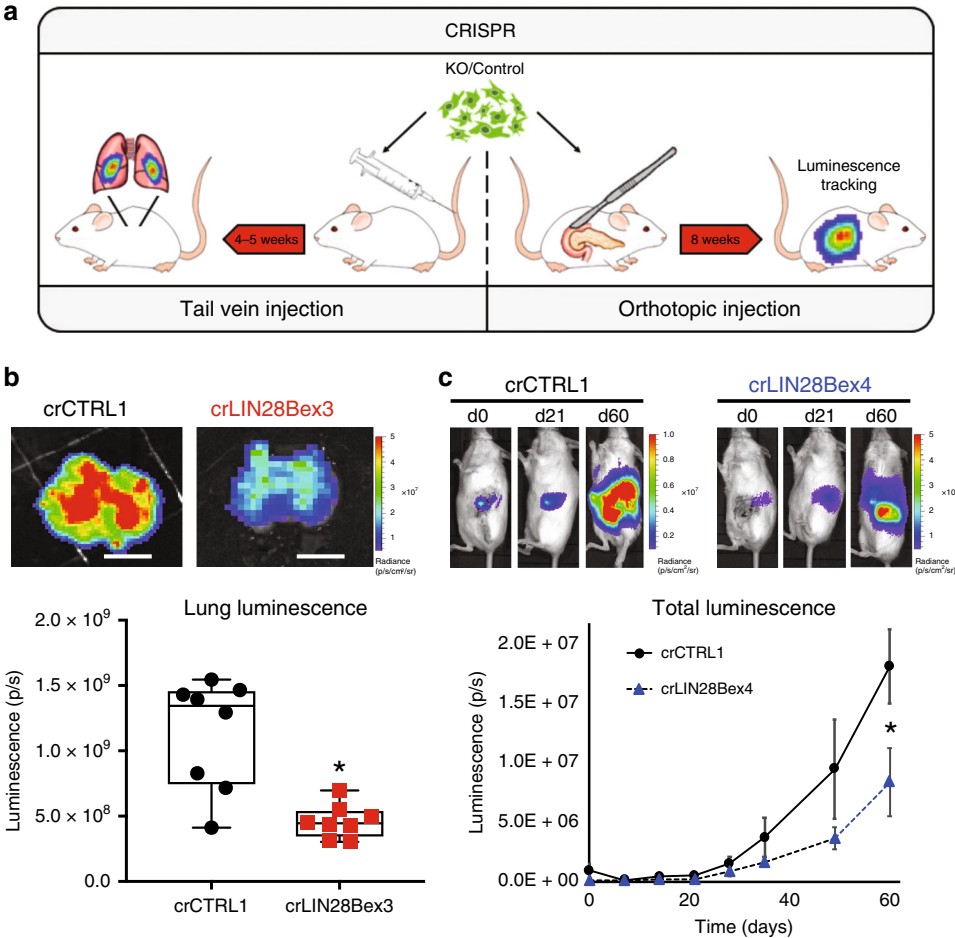

**Fig. 4 LIN28B-knockout cells are less aggressive in mouse models. a** Schematic of two complementary animal tumor metastasis models utilizing human cells placed within immunocompromised mice. **b** Representative luminescent images and quantification of lungs explanted at sacrifice from mice injected with crCTRL1 (nonsense gRNA transduced) or crLIN28B knockout (transduced with gRNA targeting LIN28B exon 3) PANC1 cells. *$p = 0.001$, 2-sided $t$-test, $n = 8$ per group, median and 25–75% IQR shown by box plots. Scale bar is 1 cm. **c** Representative time course of images acquired and quantification of whole animal bioluminescent signals after orthotopic implantation of either crCTRL1 or crLIN28B exon 4 knockout PANC1 cells. *$p = 0.035$, Mann–Whitney $U$ test, $n = 7$ per group; mean ± SEM for each time point shown.

the case with expression of genes associated with entrained leukocytes (Supplementary Fig. 4b). We also performed immunofluorescent staining and imaging on purified CTCs from an independent cohort of six patients with metastatic PDAC and confirmed heterogeneous LIN28B protein expression in the nuclei of cytokeratin-positive cells (Fig. 2e, Supplementary Fig. 5a). In our prior single-cell PDAC CTC work[11], we identified 17 CTCs collected from 4 patients (patients labeled by letter A-D, cells by addended number), all of which had the "classical" epithelial CTC phenotype as defined in that publication. We confirmed variable expression of *KRT7/8/18/19* and *MUC4/16/17* within the single CTCs, and identified significant LIN28B expression in one CTC, from that dataset (Supplementary Fig. 5b).

In addition, PDAC CTC *LIN28B* expression was statistically correlated with pancreatic developmental transcription factor *PDX1* (Supplementary Fig. 6a), a marker that has been used by other groups to identify PDAC specific CTCs[14]. To evaluate evidence of downstream *LIN28B* activity, we noted a correlation between *LIN28B* CTC expression and expression of *MIR100H*, a lncRNA shown to be co-regulated along with *LIN28B* through TGF-β signaling[27] (Supplementary Fig. 6c). Additional known let-7 targets *HMGA2* and *IGF2BP1*, but not *KRAS*, were also correlated with *LIN28B* expression (Supplementary Fig. 6d). However, we did not identify a correlation between LIN28B

expression in PDAC CTCs with *SIRT6* or *MYC* (Supplementary Fig. 6b), genes shown by others to correlate with *LIN28B* expression in murine PDAC[17]. Interestingly, the established EMT marker *FN1* was found to also correlate with *LIN28B* expression in PDAC CTCs (Supplementary Fig. 6e), whereas neither epithelial marker *CDH1* nor *EPCAM* were found to be linked with *LIN28B* expression (Supplementary Fig. 6f). To search for the origin of *LIN28B*-expressing cancer cells within primary PDAC tumors, we analyzed an independent cohort of 80 resected PDAC tumors for *LIN28B* RNA expression by RNA in situ hybridization (Supplementary Fig. 7a, Supplementary Table 5) and at the protein level by immunohistochemistry (Supplementary Fig. 7b). Consistent with previous CTC enriched markers, *LIN28B* cancer cells comprise a small subpopulation of primary tumor cells, which suggests the acquisition of *LIN288B* transcriptional programs are unique to a select number of cells with metastatic propensity. Moreover, these analyses demonstrated the inter- and intra-tumoral heterogeneity of *LIN28B* positive PDAC cells and the difficulty of developing a *LIN28B* biomarker solely from primary tumor biopsy specimens. Altogether, the association of *LIN28B* PDAC CTC expression with poor outcome and correlation with stem cell and EMT genes known to impact metastatic potential made *LIN28B* an attractive gene candidate to evaluate its functional role in the circulating and metastatic phenotypes.

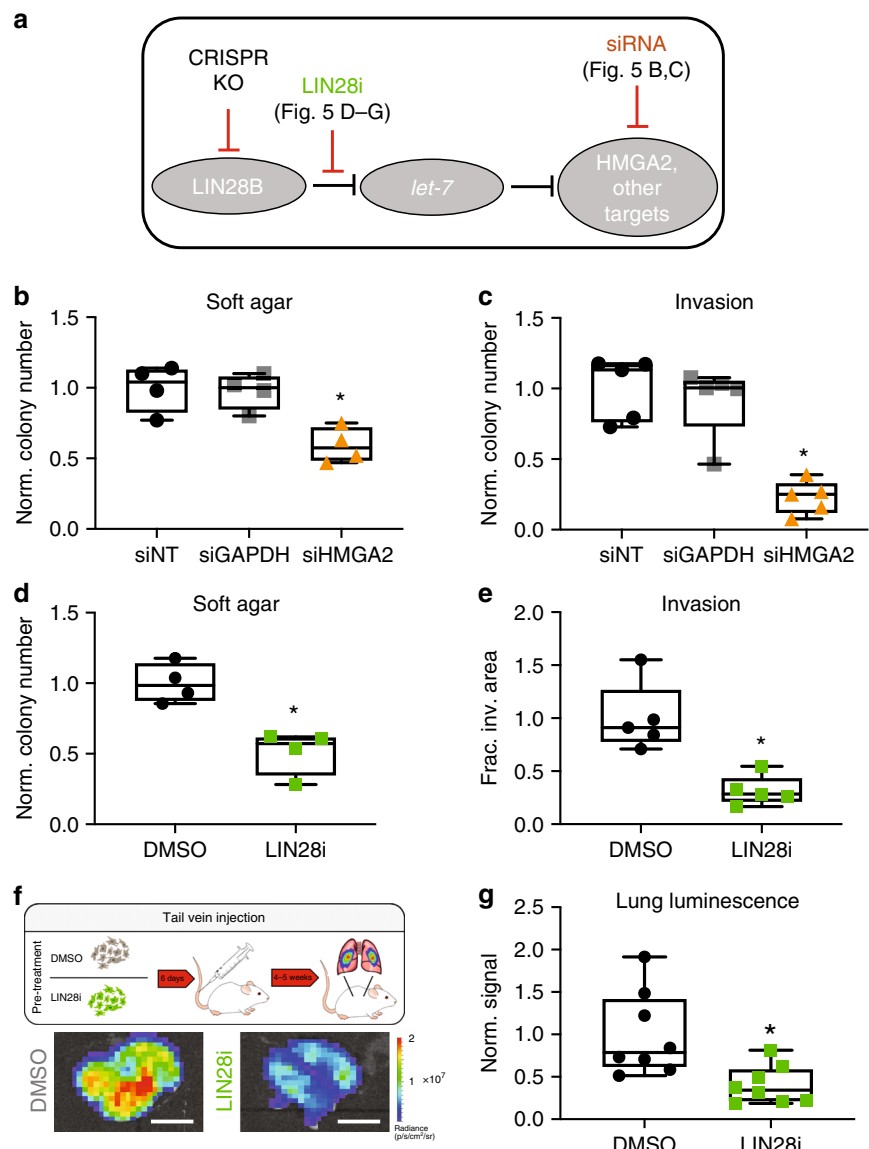

**Fig. 5 The canonical LIN28B/let-7 axis is critical for the metastatic and aggressive PDAC phenotype in vitro and in vivo. a** Schematic of LIN28B (negative) regulation of let-7 miRNA species, which then act to negatively regulate downstream mRNA targets. **b**, **c** Soft agar colony formation **b** and three-dimensional invasion assay **c** of PANC1 cells after silencing (siRNA) of HMGA2 or GAPDH compared with non-targeting siRNA control. *p < 0.05, two-tailed Bonferroni-adjusted t-test, n = 4–5 per group, median and 25–75% IQR shown by box plots. **d**, **e** Soft agar colony formation (**d**) and three-dimensional invasion assay (**e**) of PANC1 cells after treatment with 50 μM LIN28 inhibitor for 3 days prior to and then for the duration of the assay. *p < 0.005, two-tailed t-test, n = 4–5 per group, median and 25–75% IQR shown by box plots. **f**, **g** Schematic and representative images **f** with quantification of explanted lung bioluminescent signal **g** from tail vein injection metastasis model with cells pretreated for 6 days with either 0.1% DMSO (vehicle control) or 50 μM LIN28 inhibitor prior to injection. *p = 0.008, two-tailed t-test, n = 8 per group, median and 25–75% IQR shown by box plots. Scale bar is 1 cm. There were no replicates of this in vivo experiment.

**LIN28B knockout impairs PDAC metastatic phenotype**. Evaluation of a panel of PDAC cell lines identified PANC1 and PANC0327 as having high protein expression of *LIN28B* (Supplementary Fig. 8a, b). We utilized lentiCRISPRv2 constructs[28] to generate stable knockouts of *LIN28B* in these lines (Fig. 3a), which led to the expected de-repression of let-7 miRNAs compared with nonsense gRNA-transduced controls (Fig. 3b). LIN28B knockout did not impact the protein levels of KRAS, a let-7 target (Supplementary Fig. 8c) and PDAC driver, consistent with multiple regulators of KRAS expression[29].

We then identified differentially expressed genes in *LIN28B*-knockout vs control PANC1 cells by RNA-sequencing. Of the 230 genes downregulated upon LIN28B knockout, 67 were validated let-7 targets per the miRTarBase database[30] (Supplementary Fig. 8d, Supplementary Table 6). Of the top 100 downregulated genes, 32 were linked with extracellular matrix (ECM), stemness, and neural gene sets (Supplementary Fig. 8e; Supplementary Table 6). This list contained both classical let-7 targets such as *HMGA2*[16] and less-established let-7 targets such as pro-metastatic *FN1*[31], consistent with the expected and observed de-repression of let-7 species upon *LIN28B* knockout (Fig. 3b). We confirmed downregulated protein expression of LIN28B, HMGA2, and FN1—and constant expression of cytokeratins as a control—by immunofluorescent staining (Supplementary Fig. 8f, g). Interestingly, many genes that were downregulated in PANC1 cells with *LIN28B* knockout were found to be highly

correlated in the primary PDAC TCGA RNA expression dataset (Supplementary Fig. 9), suggesting an observable broader co-regulation of genes in resected primary PDAC specimens that might be attributable to LIN28B activity.

*LIN28B*-knockout PANC1 and PANC0327 cells were less aggressive in cellular assays, with a diminished capability to form nonadherent colonies in soft agar (Figs. 3c, d), migrate into two-dimensional "wounds" (Figs. 3e, f), and invade into three-dimensional extracellular matrix (ECM) gels (Fig. 3g, h). Interestingly, the ability to proliferate in two-dimensional culture was not impacted by *LIN28B* knockout (Supplementary Fig. 10), implying some specificity of *LIN28B* function on tumorigenic and metastatic potential.

*LIN28B*-knockout and control nonsense gRNA-transduced PANC1 cells co-transduced with luciferase[32] were then used for mouse tail vein inoculation and orthotopic xenograft experiments to determine changes in metastatic fitness in vivo (Fig. 4a). Tail vein injection of *LIN28B*-knockout cells resulted in significant reduction in metastatic seeding and growth in explanted lungs compared to control gRNA-transduced PANC1 cells (Fig. 4b). Consistent with these results, pancreatic orthotopic tumors had significantly less whole-body bioluminescent signal—which contained contributions from primary tumor growth, local invasion, and distant metastasis—over a 60-day experiment (Fig. 4c). Taken together, these results are consistent with a fundamental role for *LIN28B* in multiple aspects of the PDAC metastatic cascade.

**The canonical LIN28B/let-7 pathway drives PDAC metastasis.** We next sought to obtain insights into the mechanisms by which LIN28B impacts the cellular metastatic phenotype. We utilized multiple strategies to assess whether the canonical LIN28B/let-7 pathway (Fig. 5a) was dominant in the cell phenotype effects we observed with *LIN28B* knockout.

Silencing of *HMGA2* (Supplementary Fig. 11a), a classical let-7 target whose expression decreases with *LIN28B* knockout, in PANC1 cells resulted in a phenotype similar to that of *LIN28B*-knockout, with significantly reduced nonadherent colony formation (Fig. 5b) and invasion through 3D ECM gels (Fig. 5c) relative to transfection with either scrambled non-targeting siRNA (siNT) or siRNA targeting the common housekeeping gene GAPDH (siGAPDH). Transfection of *LIN28B*-knockout cells with a let-7 sponge, an oligonucleotide designed to sequester let-7 miRNAs[33] that were de-repressed in the absence of *LIN28B*, partially reversed the impaired colony formation phenotype (Supplementary Fig. 11b).

Given that genetic inhibition of both *LIN28B* and a putative dominant downstream effector (*HMGA2*) caused a similar anti-metastatic effect on PDAC cells, we hypothesized that specific chemical inhibition of *LIN28B* regulation of let-7 miRNAs would have a similar impact. Treatment of pancreatic cancer cells with a selective inhibitor of LIN28B/let-7 binding (LIN28 inhibitor: *N*-methyl-*N*-[3-(3-methyl-1,2,4-triazolo[4,3-*b*]pyridazin-6-yl)phenyl]acetamide) that has been previously shown to facilitate differentiation of embryonic stem cells via let-7 de-repression[34] also de-repressed mature let-7 expression (qPCR) in PDAC cells (Supplementary Fig. 11c). PANC1 cells treated with 50 μM LIN28 inhibitor also had a diminished ability to form nonadherent colonies (Fig. 5d, Supplementary Fig. 11d) and invasion in our cell culture model systems (Fig. 5e, Supplementary Fig. 11e). Finally, pre-treatment of PANC1 cells with 50 μM LIN28B inhibitor for six days prior to intravenous injection into mice significantly reduced the ability of the cells to seed and grow as lung metastases relative to cells treated with vehicle (Fig. 5f, g), even in the absence of in vivo drug exposure. Taken together, our data indicate that the classical LIN28B/let-7 pathway is a

dominant pathway involved in enabling certain aspects of the aggressive and metastatic PDAC cell phenotype and that disruption of key points within this pathway might serve as novel therapeutic avenues to halt metastatic dissemination.

**Discussion**

In summary, we provide a comprehensive transcriptional analysis of human PDAC CTCs that has revealed CTC correlated gene sets and identified *LIN28B* gene expression within PDAC CTCs as a prognostic factor for survival. Manipulation of the LIN28B pathway in multiple cell and animal systems confirmed a critical role for *LIN28B* in the metastatic phenotype. These findings build upon prior work by others demonstrating CTC heterogeneity for EMT[35] and stem cell[36] markers in resectable PDAC patients that correlated with differences in disease recurrence. Our unbiased characterization of CTCs purified from PDAC patients identified three major CTC subsets, a *LIN28B* and *KLF4* subset (correlated gene group 1), a *WNT5A* subset (correlated gene group 2), and an *LGALS3* subset (correlated gene group 3). These factors are linked with known EMT and stem cell pathways, and our classification system provides a new biological schema to understand metastatic behavior. Of the potential drivers we identified, we focused on *LIN28B* since it had not been previously studied specifically as a metastasis factor and since the elevated expression of *LIN28B* in PDAC CTCs correlated with poor survival in our database. CRISPR mediated knockout revealed a functional decrease in invasive and metastatic features in cell line and mouse model systems that was mediated primarily through the canonical LIN28B pathway via de-repression of let-7 miRNAs. We demonstrate that this pathway can be therapeutically targeted with a chemical inhibitor of LIN28B/let-7 interaction in both in vitro and in vivo models. This work is the first to identify *LIN28B* within human PDAC CTCs as a prognostic factor and to implicate the functional importance of *LIN28B* activity in these metastatic precursors.

Whereas our prior CTC studies primarily utilized mouse models to derive biological insights that were then validated using human biospecimens[10,11], the current work utilized primary human biospecimens as a hypothesis generating tool for subsequent characterization with model systems. Further, this is the first study to our knowledge implicating a miRNA regulatory network in CTC behavior. Importantly, our work included CTCs released from localized PDAC tumors; indeed, we and others have shown that cells can be released during early-stage (non-metastatic cancers) and even from pre-malignant cystic pancreatic lesions[14,37,38]. We found that LIN28B signaling appears to be important in CTCs derived from both clinically localized and metastatic PDAC tumors. Our study utilized bulk RNA-seq as a tool for correlated gene set definition, and although this strategy does not address single cell variability or entrainment of additional cell types, it is similar to the bulk RNA-seq technique used in other larger datasets such as the TCGA[39]. CTC-specific immunofluorescent analysis confirmed nuclear expression of LIN28B in cytokeratin-positive/CD45-negative iChip purified CTCs and a subpopulation of primary PDAC cells were found to express LIN28B at the RNA and protein level. Future development of more sophisticated computational techniques to deconvolute the effects of additional cell types as has been done in primary tumor samples[40] may be needed to better develop CTC subtyping from bulk gene expression profiling.

Notably, in our prior work utilizing primarily PDAC mouse models with advanced metastatic disease, CTCs shared expression of ECM associated genes including *FN1* and *SPARC*[10,11], implying multiple potential drivers of these pro-metastatic markers. The mechanism of LIN28B-mediated ECM gene expression

within human PDAC CTCs might be distinct from other better-characterized pathways, although TGF-β (via induction of MIR-100HG[27]) might be a common factor. Altogether, this points to a common expression program that contributes to both early and late metastatic phenotypes, but the intrinsic differences in the pathways to achieve these goals are a reflection of the cellular heterogeneity observed in CTCs and metastatic dissemination[11,26,41].

There have been several previous publications describing the association of LIN28B—usually in primary tumors—with adverse outcomes in other cancer types, including hepatocellular carcinoma[42], oral squamous cell carcinoma[43], breast cancer[44], acute myeloid leukemia[45], and multiple myeloma[46]. In addition, there seems to be a fundamental role of LIN28B in promoting the pathogenesis of MYCN-driven subsets of neuroblastoma[47]. Drivers of LIN28B expression in PDAC are incompletely described. Prior work has demonstrated the importance in PDAC of SIRT6 in MYC-driven LIN28B expression[17], but these correlations were not seen in our primary PDAC CTCs possibly due to multiple alternative pathways of LIN28B regulation. On the other hand, TGF-β is a known metastatic driver[48] and in pancreatic cancer can increase expression of both LIN28B and MIR-100HG – a lncRNA that is then processed into multiple miRNAs (mir-100, miR-125b-1, let-7a-2)[27]. Our primary CTC data showed a correlation between MIR-100HG and LIN28B, consistent with coordinate regulation of both genes by TGF-β signaling. Interestingly, statin drugs have been demonstrated to reduce inflammatory signaling and LIN28B expression[49], and this link could partially explain why statin use has been associated with reduced risk of developing PDAC[50] and with better outcomes in advanced PDAC[51,52], although there could be other LIN28B-independent mechanisms for statin effects on PDAC[53]. These hypothesis-generating observations will need further biological and clinical validation, and LIN28B CTC status could provide a blood-based biomarker that could be useful for such work.

Our results support the translational development of drugs that target the LIN28B/let-7 pathway in PDAC as a potential means to halt the dissemination of cancer cells. Given the association of LIN28B with poor outcome in CTCs collected from patients with all stage of PDAC, the therapeutic targeting of LIN28B might be relevant in both the early/localized and late/metastatic settings. More broadly, our work illustrates a general strategy to target drivers of the early spread of pancreatic cancer, perhaps after total neoadjuvant therapy (chemotherapy plus chemoradiation) and attempted surgical cure[5]. We anticipate that therapeutic targeting of the LIN28B/let-7 pathway will not be limited to PDAC and that targeting LIN28B pathway activation might impact the metastatic potential of a broad range of cancers. Our CTC platform does not rely on pre-specification of a target capture antigen such as EPCAM, and therefore it can be utilized for any cancer type. Our work provides a strategy to utilize CTC RNA expression profiling as a "liquid biopsy" to identify CTC correlated gene sets and generate potential predictive biomarkers of response to therapeutics directed against CTCs and metastasis. Indeed, there is already significant interest in the development of additional potent chemical inhibitors of the LIN28/let-7 pathway[54] that might be leveraged to generate a pipeline of drugs for PDAC and other refractory cancers. The incorporation of these novel therapies, either alone or in combination with other treatments, can provide an opportunity to prevent early metastatic tumor seeding in patients who have localized disease and provide an improved chance for curative treatment.

## Methods

**Patients and blood draws.** Human blood for CTC analysis was obtained on two existing Dana-Farber Harvard Cancer Center (DFHCC) Institutional Review Boards protocols (05–300 and 18–179) at the Massachusetts General Hospital (MGH). Blood samples from healthy donors were obtained from anonymized discarded specimens collected at a blood donation center. Patients were consented and enrolled prior to blood draws. From 05–300, a maximum of 20 mL of blood was collected from a total of 56 subjects, of which 21 were healthy donors, 17 had localized PDAC, and 18 had metastatic PDAC. Patient cohorts and clinical characteristics are provided in Tables 1 and 2 and Supplementary Table 1. For DFHCC 18–179 (NCT03563248), the blood draw parameters were the same, and the patients all had localized PDAC; since this trial remains ongoing, we did not have access to clinical and/or outcome data from these patients for the purposes of the current publication. Patient allocation to each the groups noted in our work was not random and was defined by their disease state. Blinding during collection and analysis was not performed, as knowledge of each "type" of patient (HD, locPDAC, metPDAC) group was necessary for the analysis.

**Ctc enrichment.** CTC enrichment was performed by leucocyte depletion using the microfluidic CTC-iChip. Prior to processing the blood, the total WBC count was determined by using cell blood count machine. Next, the blood samples were incubated with anti-human CD45 antibody (clone 2D1, R&D Systems, BAM1430) and anti-human CD66b antibody (Abd Serotec, 80H3) at 100 fg/WBC and 37.5 fg/WBC. After an incubation time of 20 min rocking at room temperature, Dynabeads MyOne Strepavidin T1 magnetic beads (Life Technologies, 65602) were added and incubated rocking at room temperature for an additional 20 min. The blood was then loaded onto the CTC-iCHIP. After a 5 min centrifugation at $5200 \times g$ the CTC-enriched product was resuspended in 200 μL of RNALater (Thermo Fisher Scientific), flash-frozen, and stored at −80 °C.

**RNA-sequencing and bioinformatics analysis.** For CTC-enriched products the RNeasy Plus Micro kit (Qiagen) was used. According to the manufacturer's protocol, amplified cDNA was then generated by using the SMARTer Ultra Ultra-Low-input RNA kit, version 4 (Clontech Laboratory) followed by Nextera® XT DNA Library Preparation kit (Illumina) for sample barcoding and fragmentation. Briefly, 1 μL of a 1:50,000 dilution of ERCC RNA Spike-In Mix (Thermo Fisher) was added to each sample. First-strand synthesis of RNA molecules was performed using the poly-dT-based 3'-SMART CDS primer II A (Clontech) followed by extension and template switching by the reverse transcriptase. The second strand synthesis and amplification polymerase chain reaction (PCR) was run for 18 cycles, and the amplified cDNA was purified with a 1× Agencourt AMPure XP bead cleanup (Beckman Coulter,). The Nextera XT DNA Library Preparation kit (Illumina) was used for sample barcoding and fragmentation according to the manufacturer's protocol. One nanogram of amplified cDNA was used for the enzymatic tagmentation, followed by 12 cycles of amplification and unique dual-index barcoding of individual libraries. PCR product was purified with a 1.8× Agencourt AMPure XP bead cleanup (Beckman Coulter).

For cell lines, RNA extraction from cell lines was performed by using the miRNeasy Mini kit (Qiagen). Samples were processed for RNA-sequencing using the Illumina TruSeq Total RNA-seq protocol using the RiboZero Gold depletion kit. Library construction was performed per manufacturers protocol. To validate and quantify the Libraries, a quantitative PCR using the KAPA SYBR® FAST Universal qPCR Kit (Kapa Biosystems) was performed. The individual libraries were pooled at equal concentrations, and the pool concentration was determined using the KAPA SYBR FAST Universal qPCR Kit. The pool of libraries was subsequently sequenced on a NextSeq (Illumina). The paired-end reads from the three sequencing runs were combined and aligned to the hg38 genome from the University of California, Santa Cruz (http://genome.ucsc.edu) using the STAR version 2.4.0h aligner with default settings (Alex Dobin, https://github.com/alexdobin/STAR). Reads that did not map or mapped to multiple locations were discarded. Duplicate reads were marked using the MarkDuplicates tool in picard-tools-1.8.4 (Broad Institute, broadinstitute.github.io/picard) and were removed. The uniquely aligned reads were counted using htseq-count in the intersection-strict mode against the Homo_sapiens.GRCh38.79.gtf annotation table from Ensembl (Hinton, Cambridge, United Kingdom, www.ensembl.org). Data were then imported into the R statistical programming language for analysis (DESeq2) and heat map generation (median-normalized log 10-tranformed reads per million plus 1).

**Statistical analysis.** Analysis of parametric data was performed utilizing Graph-pad Prism software (v8) or Microsoft Excel (v15). Comparisons between two groups were made utilizing a two-tailed t-tests. Comparisons between three or more groups were made utilizing Bonferonni-adjusted two-tailed tests after one-way ANOVA calculations. Analysis of non-parametric data was performed using Graphpad prism software (v8). Kaplan–Meier survival analysis via the Gehan–Breslow–Wilcoxon method was performed using Graphpad prism software v8. A $p$-value of <0.05 using the appropriate statistical test was considered significant.

**Reporting summary.** Further information on research design is available in the Nature Research Reporting Summary linked to this article.

## Data availability

The RNAseq data have been deposited with National Center for Biotechnology Information Gene Expression Omnibus under the accession number GSE144561. All primary numerical data are provided in a Supplementary "SourceData.xlsx" file. The TCGA data referenced in the study were accessed from Cbioportal[55,56]. The Cancer Cell Line Encyclopedia was used MirTarBase version 7.0[30] was used for miRNA target identification. A Reporting Summary for this article is available as a Supplementary Information file. Source data are provided with this paper.

## Code availability

All computer code is available upon request. Source data are provided with this paper.

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

## Acknowledgements
We thank all laboratory members for helpful discussions. This work was supported by the Howard Hughes Medical Institute (D.A.H.), the National Foundation for Cancer Research (D.A.H.), National Institutes of Health (NIH) Grant R01CA129933 (D.A.H.), 2U01EB012493 (D.A.H.), and R01CA235412 (D.T.T.), the Burroughs Wellcome Fund (D.T.T.), the Warshaw Institute for Pancreatic Cancer Research (D.T.T.), the Verville Family Pancreatic Cancer Research Fund (D.T.T.), the NSF PHY-1549535 (D.T.T.), SU2C and Lustgarten Foundation (T.S.H., D.P.R., D.T.T.), the Susan G. Komen for the Cure Grant KG09042 (to S.M.), and the NCI Federal Share Program and Income (S.M.).

## Author contributions
Conceptualization by J.W.F. and D.T.T. Data curation by J.W.F. and J.P. Formal analysis by J.W.F. and J.P. Investigation by J.W.F., J.P., P.M., I.B., A.L., C.Y., E.T., H.Z., M.L., B.N., E.M.T., N.D., A.S.K., and A.S. Methodology by J.W.F., J.P., and P.M. Resources by J.W.F., J.P., P.M., A.S.L., C.F.C., T.S.H., D.P.R., S.M., D.A.H. and G.Q.D. Supervision by S.M., D.A.H., G.Q.D. and D.T.T. Visualization by J.W.F. and J.P. Writing by J.W.F., J.P. and D.T.T.

## Competing interests
D.T.T. has received consulting fees from Merrimack Pharmaceuticals, Ventana Roche, Foundation Medicine, Inc., and EMD Millipore Sigma, which are not related to this work. D.T.T. is a founder and has equity in TellBio, Inc. that is related to circulating tumor cells and with intellectual property that is to be licensed to the company. D.T.T. is a founder and has equity in ROME therapeutics and PanTher Therapeutics, which is not related to this work. D.T.T. receives research support from ACD-Biotechne, PureTech Health, and Ribon Therapeutics, which were not related to this work. Dr. Ting's interests were reviewed and are managed by Massachusetts General Hospital and Mass General Brigham in accordance with their conflict of interest policies.
