## [Peer Review File · Nature Communications]

Reviewers' comments:

Reviewer #1 (Remarks to the Author):

This is an important study by Franses et al in identifying epigenetic drivers of metastatic propensity in PDAC CTCs. Importantly, the authors have used CTCs isolated from patient samples with localized and metastatic disease (compared with healthy donors) to identify transcriptional networks associated with metastases. This has led to elucidation of LIN28B/let-7 axis as deregulated in PDAC CTCs from both localized and metastatic PDAC. Inhibition of LIN28 function - either genetic, or pharmacological through disruption of LIN28/let-7 interaction - downregulates metastatic propensity in preclinical models.

Overall this is an important study using bona fide patient derived CTCs and functional studies that provide compelling evidence vis-a-vis the relevance of this axis in PDAC metastases.

The only suggestion that I hope the authors can address are RNA in situ studies using PDAC tissues confirming the existence of (?heterogeneous) LIN28 expressing cell populations (and if there is any group that can feasibly pull RNA in situ studies off, it is this team of authors). Are these LIN28B high cellular sub-populations (presumably source of the corresponding CTCs) readily detectable in localized PDAC? Are they topographically present at so-called "leading edge" of the tumors or at/near sites of lymphovascular invasion? Is there a correlation of the frequency of detection with subsequent metastatic progression?

Getting access to metastatic PDAC samples is a logistical challenge so this is not a mandatory "ask" but it would also be nice to know if there is an enrichment for LIN28B expressing populations at metastatic sites.

Reviewer #2 (Remarks to the Author):

The authors are motivated to better understand drivers of early recurrence and identify therapeutic targets to inhibit metastatic dissemination in PDAC. They purify circulating tumor cells from human PDAC cases (17 localized, 18 metastatic) and compare those to 21 healthy controls. The authors have extensive experience with the CTC-iChip for CTC purification, which is based on depletion of other hematopoietic cells, not relying on cell-surface markers of CTCs. From bulk RNA sequencing data of CTCs collected in these patients, they identify gene signatures enriched in the cancer patients compared to controls, focusing on keratin and mucin genes as evidence for CTC detection. They

identify 3 correlated gene sets within the genes enriched in PDAC CTCs, and show enrichment for stemness genes in these sets. The remainder of the study focuses in LIN28B, its target let-7, and additional downstream targets, which they show to have some relationship with patient survival. In vitro experiments on CRISPR knockout of LIN28B and investigation of metastatic potential of cells with a LIN28B genetic or chemical silencing in a mouse model demonstrate the potential of LIN28B/let-7 and its downstream network as a therapeutic target in PDAC.

The work as presented lacks some novelty, as the authors have previously published data on CTCs isolated from human PDAC patients, but the downstream analysis of LIN28B is interesting and will likely contribute to the field. I have some major concerns regarding the analysis of and claims made from the RNA sequencing data which are described below:

Major comments:

1. The results presented in Figure 1 on keratin and mucin gene expression in CTCs is somewhat unconvincing. The results would be strengthened by analysis of these genes in other relevant PDAC datasets.

a. For example, how do these 9 genes behave in the Moffitt 2015 Nature Genetics dataset, which includes primary, metastatic, and normal pancreas samples? Are they specific to tumor samples or present in normal pancreas as well?

b. How were the KRT and MUC genes shown here selected? At least a reference if not data supporting that choice from other public datasets should be included. Why not include other CTC gene markers, i.e. SPARC, identified in previous studies?

c. Why do the authors use the KRT gene expression as part of their metasignature of CTC detection, when the KRT sum shown in S1A shows a large overlap between healthy and cancer patients?

d. The authors claim that this metasignature represents CTC detection. Does it correlate with patient outcome? Can it be validated in any other CTC datasets such as those in the authors own papers (Yu et al 2012 and Ting et al 2014)?

2. The presentation of the 3 “subtypes”, or correlated sets of genes, that the authors identify also needs some improvement and external validation.

a. The gene sets that makeup these “subtypes” should be presented in a standard gene by sample heatmap. The current correlation plot does not demonstrate how these genes look across samples. The claim of “subtype” should not be made without demonstrating that these gene signatures are mutually exclusive across patients.

b. How do these subgroups of genes behave in other PDAC tumor datasets? How do they relate to PDAC subtypes previously described (i.e. Collison 2011, Moffitt 2015, Bailey 2016, Puleo 2018)?

c. The CTC populations used for bulk sequencing are likely a mixture of cell types, including true PDAC CTCs but possibly other confounding cell types as well. Perhaps investigation of these gene sets in the Ting et al 2014 single cell data would support the assertion that these gene sets represent different programs within the true PDAC CTCs.

3. Supplementary Figure S2 shows some surprising results: LIN28B, KLF4, and WNT5A are generally more highly expressed in the CTCs of localized patients compared to metastatic patients. Yet the authors show that LIN28B high expression correlates with worse patient outcome. Can the authors comment on this result? Additionally, it is not clear if the survival plots in 2C are based on the single genes labeled, or on a signature from the clusters of genes shown.

4. It is not clear if the survival analysis shown in Figure S3 on the TCGA dataset uses the correct set of annotated cases (verified PDAC vs. normal or neuroendocrine). For example, the analysis of microRNAs from the PROGmiR tool includes 191 cases, which is more cases than in the entire TCGA PDAC dataset (N=185 according to Genomic Data Commons Portal). The authors should cite the original TCGA PDAC publication and repeat these survival analyses using data from the Genomic Data Commons Portal filtered appropriately for PDAC only, referring to the original publication for details. Furthermore, why are different genes shown in Kaplan-Meier plots for TCGA data and for the primary dataset— why not treat the TCGA data as a validation set for what is found in this CTC dataset?

5. CTCs isolated with the CTC-iChip may be contaminated by leukocytes or other circulating epithelial cells (as demonstrated in the healthy controls), as the authors have noted in previous studies. Bulk RNA sequencing does not resolve this heterogeneity. This particularly becomes an issue when trying to identify de novo subtypes across patients, as the patient to patient variability may be driven by technical differences in purity of the population. The most obvious way to address this issue is to look for tumor-specific mutations in the supposed CTCs. This information is potentially available in the existing data, since mutations can be identified in bulk RNA sequencing data. Even without data from corresponding primary tumors, the authors could look for the canonical KRAS mutations present in the vast majority of PDAC cases. More rigorously, mutation analysis of rare variants in known PDAC mutated genes could provide convincing evidence that the bulk CTC population does in fact represent cells derived from the tumor, and demonstrate the level of purity of the CTCs. Without this, readers may be left skeptical of what the bulk RNA sequencing data represents.

Minor comments:

1. Color keys are needed for heatmaps in Figures 1 and 2.

2. Presumably Figure 1D contains a Venn diagram but the circles are missing in the version I received for review.
3. The number of patients in each group for the 2C Kaplan-Meier curves should be shown in the figure or listed in the figure legend.
4. The description of the patient cohort in the methods section is not consistent with the cohorts presented in the results (N=35 vs. N=56).
5. It would be helpful for readers to know exactly, or at least approximately, how many single cells were isolated using the CTC-iChip for each patient sample.
6. The authors state that RNA sequencing data will be submitted to GEO, but they should provide a GEO accession number.
7. Supplemental tables with gene lists should be presented in Excel tables rather than PDF form, which would greatly improve the accessibility of the results to readers.

Reviewer #3 (Remarks to the Author):

Pancreatic Circulating Tumour Cell Profiling Identifies LIN28B as a Metastasis Driver and Drug Target.
Franses JW et al.

CTCs of 35 PDAC patients with locoregional (n=17) and metastatic (n=18) disease have been isolated using the CTC-iChip developed by Toner & Haber. Upon RNA-sequencing profiling CTC three main CTC-subtype clusters have been identified with respect to stemness-related genes, i.e. LIN28B and KLF4, WNT5A, and LGALS3. These clusters were not exclusive to early or late-stage PDAC CTCs. LIN28B was the most important candidate as high expression in CTCs correlated with worsened survival in the patient cohort. Moreover LIN28B expression correlated with EMT transcription markers, implying a potential role in disease dissemination through the blood circulation.

To prove a functional role in the metastatic process of LIN28B in PDAC, the authors have created a CRISPR-mediated knockout in pancreatic cancer cell lines PANC1 and PANC0327, and found that let-7 miRNAs were de-repressed, i.e. the metastatic phenotype was impaired. RNA-sequencing analysis was carried out in knockout cells as well, along with cell culture assays for adherence, invasion and migration, revealing a less aggressive phenotype of the LIN28B knockout cell lines.

LIN28B-knockout and control nonsense gRNA-transduced PANC1 cells co-transduced with luciferase were then tested in vivo by mouse tail vein inoculation and orthotopic xenograft experiments in order to determine changes in metastatic ability. Tail vein injection of LIN28B

knockout cells revealed a significantly reduced metastatic process compared to control gRNA-transduced PANC1 cells, while pancreatic orthotopic tumours had significantly less total animal bioluminescent signal.

siRNA-based assays were applied to identify the molecular pathways behind the observations. The experiments implied that inhibition of LIN28B regulation of let-7 miRNAs may exert an anti-metastatic effect on PDAC cells similar to LIN28B and downstream HMGA2 inhibition.

The authors conclude that the LIN28B/let-7 pathway is a dominant pathway involved in the aggressive / metastatic PDAC cell phenotype, and that disruption within this pathway may serve as future targets for therapeutic intervention of metastatic disease.

The experimental work is well performed and the authors have illustrated their findings with supportive tables and detailed figures. As little has been known about the influence of LIN28B on metastatic behavior in PDAC cells before, this work contributes novel information relevant to a better understanding of cancer metastasis in pancreatic cancer.

Comments:

- The subgroups of patients with locoregional and distant metastatic spread are rather small and they are mixed in the Kaplan-Meier survival analyses, which limits the significance of this analysis. A further validation on a larger independent cohort is recommended.
- The authors state that their CTC markers are pancreas-specific (page 5) but according to my best knowledge MUC expression is also found in normal blood cells and normal epithelial cells.
- Were the HD controls age-matched with the cancer patients?
- The present data do not support the statement that “these findings confirm that entry into circulation can be an early event”. The authors have not analyzed any patient with early pancreatic cancer (e.g., pT1 tumors, node-negative).
- Survival analysis for LIN28B high PDAC CTCs has only been presented in univariate analysis. Has a multivariate analysis been calculated including confounding clinical parameters?
- Instead of analyzing three or more groups by one-way ANOVA followed by t-tests, the authors have to use one-way ANOVA with post-hoc Tukey’s HSD test.
- Figure 5: single measurements in the boxplot for siGAPDH are missing.

Reviewer #1 (Remarks to the Author):

This is an important study by Franes et al in identifying epigenetic drivers of metastatic propensity in PDAC CTCs. Importantly, the authors have used CTCs isolated from patient samples with localized and metastatic disease (compared with healthy donors) to identify transcriptional networks associated with metastases. This has led to elucidation of LIN28B/let-7 axis as deregulated in PDAC CTCs from both localized and metastatic PDAC. Inhibition of LIN28 function - either genetic, or pharmacological through disruption of LIN28/let-7 interaction - downregulates metastatic propensity in preclinical models. Overall this is an important study using bona fide patient derived CTCs and functional studies that provide compelling evidence vis-a-vis the relevance of this axis in PDAC metastases.

1.1. The only suggestion that I hope the authors can address are RNA in situ studies using PDAC tissues confirming the existence of (?heterogeneous) LIN28 expressing cell populations (and if there is any group that can feasibly pull RNA in situ studies off, it is this team of authors). Are these LIN28B high cellular sub-populations (presumably source of the corresponding CTCs) readily detectable in localized PDAC? Are they topographically present at so-called "leading edge" of the tumors or at/near sites of lymphovascular invasion? Is there a correlation of the frequency of detection with subsequent metastatic progression? Getting access to metastatic PDAC samples is a logistical challenge so this is not a mandatory "ask" but it would also be nice to know if there is an enrichment for LIN28B expressing populations at metastatic sites.

We thank Reviewer #1 for the useful critique and encouraging comments. To examine these issues, we performed RNA in-situ hybridization (RNA-ISH) staining of human primary PDAC tumor microarrays (TMAs) to attempt to identify enriched expressions in particular subpopulations of glandular pancreatic cancer cells in a cohort of patients. We found heterogeneous expression of LIN28B transcripts – both between patients and between glands within a patient's specimen. These types of heterogeneity are consistent with the gland-to-gland variability that our group published recently¹. We have inserted a new supplementary subfigure (Fig. S6) and have also inserted these data below. These new data are discussed in the revised manuscript on page 8. As you can see, there is substantial heterogeneity at both the RNA level (A below) and protein level in LIN28B (B below). This inter-tumor and intra-tumor heterogeneity is consistent with our recent findings showing gland-to-gland variability in epithelial and quasi-mesenchymal phenotypes¹.

A**B**
When we scored an independent cohort of 80 resected localized PDAC tumors (as part of an existing tumor tissue microarray set) for *LIN28B* expression by RNA-ISH (Table S5 in the revised manuscript) we noted “positive” staining in 32 of 80 scorable specimens. We were unable to detect a clear enrichment of *LIN28B* expression within any specific subset of PDAC tumor glands studies and could not find a consistent location of cells within tumors (i.e. not enriched on invasive edge or near vessels).

In addition, we performed immunofluorescent staining on CTCs purified from another independent cohort of 6 patients with metastatic PDAC. We observed heterogeneous nuclear staining of *LIN28B* in cytokeratin-positive/CD45-negative cells. We inserted representative images as the new Fig. S5 in the revised manuscript (reproduced below) and also utilized a subset of the images in Fig. 2E in the revised manuscript.

Reviewer #2 (Remarks to the Author):

The authors are motivated to better understand drivers of early recurrence and identify therapeutic targets to inhibit metastatic dissemination in PDAC. They purify circulating tumor cells from human PDAC cases (17 localized, 18 metastatic) and compare those to 21 healthy controls. The authors have extensive experience with the CTC-iChip for CTC purification, which is based on depletion of other hematopoietic cells, not relying on cell-surface markers of CTCs. From bulk RNA sequencing data of CTCs collected in these patients, they identify gene signatures enriched in the cancer patients compared to controls, focusing on keratin and mucin genes as evidence for CTC detection. They identify 3 correlated gene sets within the genes enriched in PDAC CTCs, and show enrichment for stemness genes in these sets. The remainder of the study focuses in LIN28B, its target let-7, and additional downstream targets, which they show to have

some relationship with patient survival. In vitro experiments on CRISPR knockout of LIN28B and investigation of metastatic potential of cells with a LIN28B genetic or chemical silencing in a mouse model demonstrate the potential of LIN28B/let-7 and its downstream network as a therapeutic target in PDAC.

The work as presented lacks some novelty, as the authors have previously published data on CTCs isolated from human PDAC patients, but the downstream analysis of LIN28B is interesting and will likely contribute to the field. I have some major concerns regarding the analysis of and claims made from the RNA sequencing data which are described below:

Major comments:

2.1. The results presented in Figure 1 on keratin and mucin gene expression in CTCs is somewhat unconvincing. The results would be strengthened by analysis of these genes in other relevant PDAC datasets.

a. For example, how do these 9 genes behave in the Moffitt 2015 Nature Genetics dataset, which includes primary, metastatic, and normal pancreas samples? Are they specific to tumor samples or present in normal pancreas as well?

b. How were the KRT and MUC genes shown here selected? At least a reference if not data supporting that choice from other public datasets should be included. Why not include other CTC gene markers, i.e. SPARC, identified in previous studies?

c. Why do the authors use the KRT gene expression as part of their metasignature of CTC detection, when the KRT sum shown in S1A shows a large overlap between healthy and cancer patients?

d. The authors claim that this metasignature represents CTC detection. Does it correlate with patient outcome? Can it be validated in any other CTC datasets such as those in the authors own papers (Yu et al 2012 and Ting et al 2014)?

We appreciate the thoughtful feedback from Reviewer #2.

The keratin genes that we utilized are commonly used epithelial cytokeratins that encode commonly used immunohistochemical targets for identification of pancreatic cancer^{2,3,4,5,6}. Our prior publication confirmed positive but heterogeneous expression in single mouse pancreatic cancer CTCs⁷, and indeed many CTC identification strategies rely on positive staining for epithelial keratins⁸. It is important to note that these keratins are epithelial markers⁹ and are also present in normal pancreas tissue (<https://www.proteinatlas.org/ENSG00000135480-KRT7/tissue/pancreas>). We have added references (page 5, highlighted) to justify selection of these genes.

As for the mucin genes, we identified each of them (MUC3A/4/16/17) as differentially expressed genes in both locPDAC and metPDAC purified CTC populations compared to purified blood obtained from HD controls (Fig. 1C). The literature supports the notion that normal pancreas tissue does not express large quantities of most mucin genes. Indeed, examining RNA-seq data from the GTEX portal (accessed on 12/31/2019) confirm essentially no RNA expression of MUC3A, MUC4, MUC16, and MUC17 in normal pancreas tissue or whole blood:

Gene	whole blood median RPM	pancreas median RPM	highest median RPM	ID of highest median RPM site
MUC3A	0.0087	0.51	48.6	small intestine
MUC4	0.010	0.069	24.7	transverse colon

MUC16	0.0046	0.0058	4.0	minor salivary gland
MUC17	0.000	0.0019	28.1	small intestine - terminal ileum

However as pancreatic tumorigenesis progresses, multiple membrane-tethered mucins – including MUC3 and MUC4¹⁰, MUC16 (which encodes CA125)¹¹, and MUC17¹² – are upregulated. In fact, the latter MUC17 expression was found to be associated with lymph node metastasis and poor survival¹².

Given the lack of specificity of keratins and the increased specificity of mucins (Fig. S1), we felt confident that the mucin metasignature was a better readout than the keratin metasignature. We have added a comment to this effect on page 6 of the revised manuscript. Despite this, we could not find a clear correlation between the summed expression in pancreatic cancer CTCs of mucins 3A, 4, 16, and 17 and overall survival in our dataset. We have displayed these data below and added them to the revised manuscript as Fig. S1C.

In addition, we reviewed available data from our more recent prior single cell RNA-seq publication⁷ to see whether some of our genes of interest could be detected within single (via micromanipulation CTCs. There were data from 17 CTCs collected from 4 patients (patients labelled by letter A-D, cells by number), all of which had the “classical” epithelial CTC phenotype as defined in that publication. We confirmed variable expression of *KRT7/8/18/19* and *MUC4/16/17* within the single CTCs from that dataset, shown below, with RPM-normalized data being shown in heat map form with the row (per gene) maximum in red, the row minimum in blue, and the median in white. Out of these four patients we did detect concordant expression of *LIN28B* and mucin genes in one cell from a single patient. This is consistent with our results in our current manuscript, showing variable expression on a per-patient basis, of *LIN28B*. We also found variable expression of *KLF4*, *WNT5A*, and *LGALS3*, with the highest average expression for *LGALS3*. Although it would be ideal to pursue this single-cell strategy on a more comprehensive scale, due to cost constraints we are unable to fund this effort at this point in time.

As an orthogonal assay, we also did immunofluorescence for protein expression of KRTs as is the standard CTC assay used in the field from a separate cohort of 6 patients with metastatic PDAC. We observed heterogeneous nuclear staining of LIN28B in cytokeratin-positive/CD45-negative cells. We inserted representative images as the new Fig. S5 in the revised manuscript (reproduced below) and also utilized a subset of the images in Fig. 2E in the revised manuscript.

Finally, we wanted to examine the heterogeneity of KRT and MUC gene expression in an independent dataset. Toward that end, we downloaded the histology-verified (n = 179) pancreatic ductal carcinoma bulk RNA seq data (Z-score normalized) from Cbioportal and heatmapped the KRT and MUC gene expression Z scores. As we had anticipated, and consistent with our current data and past publications, there was significant variability in the expression of both epithelial keratin genes (*KRT7/8/18/19/23*) and mucin genes (*MUC4/16/17*; N.B. there were no data available for *MUC3A*) across the dataset of 179 patients with RNA-seq data. We have shown this heatmap here and have added it into the supplement as Fig. S1D; we have also added additional text explaining this addition on page 6 of the revised manuscript.

TCGA PDAC (n = 179)

2.2. The presentation of the 3 “subtypes”, or correlated sets of genes, that the authors identify also needs some improvement and external validation.

a. The gene sets that makeup these “subtypes” should be presented in a standard gene by sample heatmap. The current correlation plot does not demonstrate how these genes look across samples. The claim of “subtype” should not be made without demonstrating that these gene signatures are mutually exclusive across patients.

b. How do these subgroups of genes behave in other PDAC tumor datasets? How do they relate to PDAC subtypes previously described (i.e. Collison 2011, Moffitt 2015, Bailey 2016, Puleo 2018)?

c. The CTC populations used for bulk sequencing are likely a mixture of cell types, including true PDAC CTCs but possibly other confounding cell types as well. Perhaps investigation of these gene sets in the Ting et al 2014 single cell data would support the assertion that these gene sets represent different programs within the true PDAC CTCs.

We appreciate again the detailed and thoughtful comments and questions.

We have replaced the heatmap in Fig. S4B to include the four stemness genes on which we focused (*LIN28B*, *KLF4*, *WNT5A*, *LGALS3*), epithelial and mucinous genes, and leukocyte genes. We have inserted it below as well. The revised heatmap (units of $\log_{10}(\text{RPM}+1)$, median centered) shows how LIN28B-high samples (e.g. locPDAC01-02, metPDAC15-18) also have high expression of mucins and variable expression of keratins. There is no relationship between high expression of leukocyte markers such as *PTPRC* (CD45) - that are presumed to be expressed by entrained / co-purified white blood cells – and LIN28B. These findings, along with the established correlation of LIN28B expression with the sum of mucin expression (Fig. 2D), support our contention that LIN28B expression is occurring within the CTCs.

Additionally, as noted above in response to the comment from Reviewer #1, we performed RNA-ISH and IHC to confirm expression of *LIN28B* transcript and LIN28B protein within cancer cells in PDAC primary tumors and immunofluorescent staining of PDAC CTCs to confirm LIN28B nuclear protein expression within cytokeratin-positive purified CTCs. Please see our response above for more detail.

We also added a new supplementary figure, which is now the new Fig. S2 in the revised manuscript, to address further some of these concerns. First, we heatmapped in our existing dataset the top 20 genes correlated with each of the 4 stemness genes on which we focused (Fig. S2A). We have commented on this result on page 7 of the revised manuscript. As requested by the reviewer, the genes found in the correlation matrix analysis were indeed well correlated with the stemness genes, as made clear with our heatmap. In addition, we utilized data from CTCs being collected from an independent cohort of 25 treatment-naïve patients at our institution who are participating on a pancreatic cancer neoadjuvant trial for borderline and locally advanced unresectable pancreatic cancer. We do not yet have access to the clinical data from the trial, but we have been allowed to provisionally analyze the data taken prior to initiation of systemic therapy. As you can see below and in the revised **Fig. S2B and S4B**, genes that we found in our original analysis that were correlated with the stemness factors (*LIN28B*, *KLF4*, *WNT5A*, *LGALS3*) were again correlated with the same stemness factors in the independent dataset.

New Fig. S2

We also performed per-sample hierarchical clustering analysis of our 35 PDAC (17 locPDAC and 18 metPDAC) CTC samples using the top 20 genes that correlated with each of *LIN28B*, *LGALS3*, *WNT5A*, and *KLF4* (and also included the 4 stemness genes as well; total of 84 genes used for clustering) and showed that, indeed, *LGALS3*-high samples (left-most box) clustered

separately from *WNT5A*-high samples (middle box) and *LIN28B*-high (right-most box) samples. This is a new Fig. 2B and we also show the data below.

(new fig. 2B)

We then utilized the Cbioportal resource to download and map the expression of these groups of genes in the TCGA histology-verified pancreatic ductal carcinoma bulk tissue Z score-normalized RNA seq dataset. We were not surprised to find that a lack of clear gene expression subgroups containing the same genes as were correlated within our CTC data. We have inserted the heatmap for these data below and in the revised Fig. S3C, with a comment on this discrepancy on page 8 of the revised manuscript. There was a subset of ~1/4 of the data that seemed to be enriched in all of the stemness markers (LIN28B, KLF4, WNT5A, LGALS3; the left side of the heatmap that we generated). We wonder whether or not these patients had different quantities of CTCs or functionally distinct phenotypes present within the CTCs released from the primary tumors, but unfortunately these hypotheses cannot be examined for this dataset.

C**TCGA PDAC (n = 179)**
2.3. Supplementary Figure S2 shows some surprising results: LIN28B, KLF4, and WNT5A are generally more highly expressed in the CTCs of localized patients compared to metastatic patients. Yet the authors show that LIN28B high expression correlates with worse patient outcome. Can the authors comment on this result? Additionally, it is not clear if the survival plots in 2C are based on the single genes labeled, or on a signature from the clusters of genes shown.

We appreciate these observations. Although the proportions of patients with high LIN28B CTC expression is numerically higher in the locPDAC group compared with the metPDAC group, a simple Chi-squared test shows that the proportions are not significantly different. With greater numbers of patients in subsequent follow-up studies, perhaps this numerical difference will be borne out, but that remains to be seen. Although in the old Fig. S2B (new Fig. S4B) the median LIN28B expression in the locPDAC group is higher than that in the metPDAC group, this more likely is also related to the numerically higher proportion of “positive” or “high” values and fewer zeros.

In our survival analyses – which took the CTC bulk RNA expression of each single gene that was labelled – we grouped all of the patients together because unfortunately most resected pancreatic cancers recur, with a 90% chance of recurrence after resection for even N1 disease¹³. We used single gene expression thresholds of the highest quartile of expression for each of the listed single genes, which for LIN28B was 16.5 RPM; we inserted a comment to this effect on page 8 of the revised manuscript. This also explains why the numbers in each “high” and “low” subgroup are the same in each of the three survival curves in the revised Fig. 2B.

2.4. It is not clear if the survival analysis shown in Figure S3 on the TCGA dataset uses the correct set of annotated cases (verified PDAC vs. normal or neuroendocrine). For example, the analysis of microRNAs from the PROGmiR tool includes 191 cases, which is more cases than in the entire TCGA PDAC dataset (N=185 according to Genomic Data

Commons Portal). The authors should cite the original TCGA PDAC publication and repeat these survival analyses using data from the Genomic Data Commons Portal filtered appropriately for PDAC only, referring to the original publication for details. Furthermore, why are different genes shown in Kaplan-Meier plots for TCGA data and for the primary dataset– why not treat the TCGA data as a validation set for what is found in this CTC dataset?

We appreciate these series of questions and comments.

The reason that different genes were shown in Kaplan-Meier plots for TCGA data versus the primary data was to emphasize that – in an independent data set – that alternative markers of LIN28B pathway activation in primary resected PDAC tissue were correlated with a poor outcome. This lent indirect but concordant support to our contention that LIN28B expression in circulating/metastatic PDAC cells is a poor prognostic factor.

In addition, in order to assess in a more clearly histologically defined population of bona fide PDAC, we utilized the heat map function within Cbioportal to look for markers of LIN28B pathway activation in histologically annotated pancreatic ductal adenocarcinoma cases from the 2017 TCGA study ¹⁴. Utilizing the heatmap functionality within Cbioportal, we found that expression of the *IGF2BP1/2/3* and *HMGA2* (canonical let-7 targets that should therefore correlate with *LIN28B* expression) genes were higher in two of four transcriptionally defined subgroups by hierarchical clustering. We show these data below and have now inserted a revised Fig. S3A to include these.

A similar examination of the TCGA data utilizing Cbioportal showed an enrichment of high expression of both *KLF4* and *LGALS3* – but not *WNT5A* – within 2 of the transcriptional subtypes below.

We then performed our own survival analysis utilizing *HMGA2* expression in the TCGA “PAAD” dataset. This dataset included 179 cases of histologically verified adenocarcinoma for which RNA seq data was also available. We confirmed that high expression of *HMGA2* (defined as positive Z score for RNA seq performed in in bulk tissue-extracted RNA) was associated with statistically significant worsened median overall survival. We show this figure below and have added it to the revised manuscript as Fig. S3B.

2.5. CTCs isolated with the CTC-iChip may be contaminated by leukocytes or other circulating epithelial cells (as demonstrated in the healthy controls), as the authors have noted in previous studies. Bulk RNA sequencing does not resolve this heterogeneity. This particularly becomes an issue when trying to identify de novo subtypes across patients, as the patient to patient variability may be driven by technical differences in purity of the population. The most obvious way to address this issue is to look for tumor-specific mutations in the supposed CTCs. This information is potentially available in the existing data, since mutations can be identified in bulk RNA sequencing data. Even without data from corresponding primary tumors, the authors could look for the canonical *KRAS* mutations present in the vast majority of PDAC cases. More rigorously, mutation analysis of rare variants in known PDAC mutated genes could provide convincing evidence that the bulk CTC population does in fact represent cells derived from the tumor, and demonstrate the level of purity of the CTCs. Without this, readers may be left skeptical of what the bulk RNA sequencing data represents.

We thank the reviewer for this important technical comment. Although the purification factor of the CTC iChip-processed mononuclear cell population is 10^4 or higher¹⁵, the iChip output is not pure. In addition, since RNA isolation is a destructive technique and we strive to maximize the signal, we do not routinely remove aliquots for alternative assays such as cytological enumeration. In prior studies the numbers of CTCs in pancreatic cancer patients had a median of 10 cells per mL of whole blood¹⁶.

The reviewer is correct in observing that the utilization of bulk RNA sequencing cannot provide single cell resolution for gene expression profiling of purified circulating tumor cells. However, TCGA and ICGC RNA-seq data also utilize bulk tissue expression data, and analysis of these data can still yield useful classification schemes. In our revised Fig. S4B (reproduced again below) we did not see a clear relationship between expression of leukocyte marker genes such as *PTPRC* (CD45) or others with *LIN28B* expression in our bulk PDAC CTC data. On the contrary, *LIN28B* was well-correlated with mucin genes and keratin genes.

We have addressed this issue as a comment in the discussion on page 13 of the revised manuscript. There are technically and computationally complex methods such as virtual microdissection by Moffitt et al¹⁷, and these techniques can be useful in improving the signal to noise ratio. However, we do not yet have the expertise to perform these sorts of analyses. We would eagerly look forward to working with collaborators who have these types of refined tools in order to refine our existing and future work.

Finally, although we did not perform a new comprehensive or quantitative analysis, we did observe significant nuclear expression of *LIN28B* protein in cytokeratin-positive/CD45-negative CTC iChip-purified cells from PDAC patients. We showed those data in our response to reviewer #1 above and have added them to the revised manuscript as a new Fig. 2E and Fig. S5.

Minor comments:

1. Color keys are needed for heatmaps in Figures 1 and 2.
2. Presumably Figure 1D contains a Venn diagram but the circles are missing in the version I received for review.
3. The number of patients in each group for the 2C Kaplan-Meier curves should be shown in the figure or listed in the figure legend.

4. The description of the patient cohort in the methods section is not consistent with the cohorts presented in the results (N=35 vs. N=56).
5. It would be helpful for readers to know exactly, or at least approximately, how many single cells were isolated using the CTC-iChip for each patient sample.
6. The authors state that RNA sequencing data will be submitted to GEO, but they should provide a GEO accession number.
7. Supplemental tables with gene lists should be presented in Excel tables rather than PDF form, which would greatly improve the accessibility of the results to readers.

Thank you very much for these detailed edits to our manuscript.

We have corrected minor comments 1-4 in the revised submission.

With respect to minor comment #5, we note above (section 2.5) that since the RNA isolation step is destructive and our goal is to maximize signal for the gene expression biomarker, we do not routinely split off aliquots of iChip-purified CTCs for internal CTC quantification/enumeration. Furthermore, in a prior publication, we found that the number of CTCs quantified by automated microscopy analysis of fixed cytopun cells was dependent on the detection technique, with fluorescent RNA in situ hybridization (RNA-ISH, with probes against KRT7/8/18/19 transcripts) being more sensitive than immunofluorescent (IF) staining (antibodies against CK8/18/19 and EpCAM)¹⁶. In that study, the median concentration of cells detected with ISH was 10 cells per mL of whole blood, but with IF it was only 4 cells/mL¹⁶.

We have recently submitted our RNA seq data to GEO and the accession number is GSE144561.

We have submitted an additional file (Tables.xlsx) containing all of the main and supplementary tables.

Reviewer #3 (Remarks to the Author):

Pancreatic Circulating Tumour Cell Profiling Identifies LIN28B as a Metastasis Driver and Drug Target. Franses JW et al.

CTCs of 35 PDAC patients with locoregional (n=17) and metastatic (n=18) disease have been isolated using the CTC-iChip developed by Toner & Haber. Upon RNA-sequencing profiling CTC three main CTC-subtype clusters have been identified with respect to stemness-related genes, i.e. LIN28B and KLF4, WNT5A, and LGALS3. These clusters were not exclusive to early or late-stage PDAC CTCs. LIN28B was the most important candidate as high expression in CTCs correlated with worsened survival in the patient cohort. Moreover LIN28B expression correlated with EMT transcription markers, implying a potential role in disease dissemination through the blood circulation.

To prove a functional role in the metastatic process of LIN28B in PDAC, the authors have created a CRISPR-mediated knockout in pancreatic cancer cell lines PANC1 and PANC0327, and found that let-7 miRNAs were de-repressed, i.e. the metastatic phenotype was impaired. RNA-sequencing analysis was carried out in knockout cells as well, along with cell culture assays for adherence, invasion and migration, revealing a less aggressive phenotype of the LIN28B knockout cell lines.

LIN28B-knockout and control nonsense gRNA-transduced PANC1 cells co-transduced with luciferase were then tested in vivo by mouse tail vein inoculation and orthotopic xenograft experiments in order to determine changes in metastatic ability. Tail vein injection of LIN28B knockout cells revealed a significantly reduced metastatic process compared to control gRNA-transduced PANC1 cells, while pancreatic orthotopic tumours had significantly less total animal bioluminescent signal.

siRNA-based assays were applied to identify the molecular pathways behind the observations. The experiments implied that inhibition of LIN28B regulation of let-7 miRNAs may exert an anti-metastatic effect on PDAC cells similar to LIN28B and downstream HMG2 inhibition. The authors conclude that the LIN28B/let-7 pathway is a dominant pathway involved in the aggressive / metastatic PDAC cell phenotype, and that disruption within this pathway may serve as future targets for therapeutic intervention of metastatic disease.

The experimental work is well performed and the authors have illustrated their findings with supportive tables and detailed figures. As little has been known about the influence of LIN28B on metastatic behavior in PDAC cells before, this work contributes novel information relevant to a better understanding of cancer metastasis in pancreatic cancer.

We thank Reviewer #3 for thoughtful assessment and summary of our initial submission.

Comments:

3.1. The subgroups of patients with locoregional and distant metastatic spread are rather small and they are mixed in the Kaplan-Meier survival analyses, which limits the significance of this analysis. A further validation on a larger independent cohort is recommended.

We agree that the survival analyses are necessarily limited by the nature of our initial study, which is hypothesis generating and needs confirmation in a larger study. As noted above, we have begun a preliminary analysis of LIN28B and the other stemness genes in CTCs purified from an independent treatment-naïve cohort of patients with locally advanced / borderline unresectable pancreatic cancer patients who are each enrolled on a neoadjuvant therapy trial (DFHCC 18-179) at the Massachusetts General Hospital Cancer Center. We observed similar heterogeneity and purity estimates (based on CD45 mRNA seq reads versus mucin reads) in the second dataset. Since the trial has not yet completed accrual and we are blinded to outcomes for data monitoring and safety, we cannot yet access the matched clinical data to assess for outcome or in fact any clinical parameter. Therefore, the clinical correlative analyses for the independent second CTC dataset cannot be completed yet and are beyond the scope of our initial study presented in this manuscript. We fully intend to explore confirmatory analyses of the LIN28B pathway in CTC and tissue sequencing from this new cohort and additional cohorts once the data become available.

3.2 The authors state that there CTC markers are pancreas-specific (page 5) but according to my best knowledge MUC expression is also found in normal blood cells and normal epithelial cells.

Thank you for the comment and thoughts.

The keratin genes that we utilized are commonly used epithelial cytokeratins that encode commonly used immunohistochemical targets for identification of pancreatic cancer^{2,3,4,5,6}. Our prior publication confirmed positive but heterogeneous expression in single mouse pancreatic cancer CTCs⁷, and indeed many CTC identification strategies rely on positive staining for epithelial keratins⁸. It is important to note that these keratins are epithelial markers⁹ and are also present in normal pancreas tissue (<https://www.proteinatlas.org/ENSG00000135480-KRT7/tissue/pancreas>). We have added references (page 5, highlighted) to justify selection of these genes.

As for the mucin genes, we identified each of them (MUC3A/4/16/17) as differentially expressed genes in both locPDAC and metPDAC purified CTC populations compared to purified blood obtained from HD controls (Fig. 1C). The literature supports the notion that normal pancreas tissue does not express large quantities of most mucin genes. Indeed, examining RNA-seq data from the GTEX portal (accessed on 12/31/2019) confirm essentially no RNA expression of MUC3A, MUC4, MUC16, and MUC17 in normal pancreas tissue or whole blood:

Gene	whole blood median RPM	pancreas median RPM	highest median RPM	ID of highest median RPM site
MUC3A	0.0087	0.51	48.6	small intestine
MUC4	0.010	0.069	24.7	transverse colon
MUC16	0.0046	0.0058	4.0	minor salivary gland
MUC17	0.000	0.0019	28.1	small intestine - terminal ileum

However as pancreatic tumorigenesis progresses, multiple membrane-tethered mucins – including MUC3 and MUC4¹⁰, MUC16 (which encodes CA125)¹¹, and MUC17¹² – are upregulated. In fact, the latter MUC17 expression was found to be associated with lymph node metastasis and poor survival¹².

Given the lack of specificity of keratins and the increased specificity of mucins (Fig. S1), we felt confident that the mucin metasignature was a better readout than the keratin metasignature. We have added a comment to this effect on page 6 of the revised manuscript. Despite this, we could not find a clear correlation between the summed expression in pancreatic cancer CTCs of mucins 3A, 4, 16, and 17 and overall survival in our dataset. We have displayed these data below and added them to the revised manuscript as Fig. S1C.

OS for all patients, split by median

In addition, we reviewed available data from our more recent prior single cell RNA-seq publication⁷ to see whether some of our genes of interest could be detected within single (via micromanipulation CTCs. There were data from 17 CTCs collected from 4 patients (patients labelled by letter A-D, cells by number), all of which had the “classical” epithelial CTC phenotype as defined in that publication. We confirmed variable expression of *KRT7/8/18/19* and *MUC4/16/17* within the single CTCs from that dataset, shown below, with RPM-normalized data being shown in heat map form with the row (per gene) maximum in red, the row minimum in blue, and the median in white. Out of these four patients we did detect concordant expression of *LIN28B* and mucin genes in one cell from a single patient. This is consistent with our results in our current manuscript, showing variable expression on a per-patient basis, of *LIN28B*. We also found variable expression of *KLF4*, *WNT5A*, and *LGALS3*, with the highest average expression for *LGALS3*. Although it would be ideal to pursue this single-cell strategy on a more comprehensive scale, due to cost constraints we are unable to fund this effort at this point in time.

As an orthogonal assay, we also did immunofluorescence for protein expression of KRTs as is the standard CTC assay used in the field from a separate cohort of 6 patients with metastatic PDAC. We observed heterogeneous nuclear staining of *LIN28B* in cytokeratin-positive/CD45-negative cells. We inserted representative images as the new Fig. S5 in the revised manuscript (reproduced below) and also utilized a subset of the images in Fig. 2E in the revised manuscript.

3.3. Were the HD controls age-matched with the cancer patients?

The HD controls were not completely age-matched with the cancer patients, with average ages of 63.9 (metastatic PDAC cohort), 68.8 (localized PDAC cohort), and 45.5 (HD cohort). We acknowledge that this could mask subtle age-related differences in the entrained leukocytes present in the purified iChip product. However, given that both localized PDAC and pre-neoplastic intraductal papillary mucinous neoplasms (IPMNs) of the pancreas can release cells into the circulation¹⁶ and that these malignant and benign tumors are associated with advanced age, we wanted to make the cleanest comparison and hence we elected purposefully to choose younger healthy patients with the lowest possible risk for harboring occult precancerous lesions.

3.4. The present data do not support the statement that “these findings confirm that entry into circulation can be an early event”. The authors have not analyzed any patient with early pancreatic cancer (e.g., pT1 tumors, node-negative).

We thank the reviewer for the comment. However, we emphasize that 4 of the 17 “locPDAC” patients had T1 disease and 11 of the 17 had N0 disease. 9 of the 17 “locPDAC” patients had TNM stage 1 disease as is noted in the last column of Table 1. In addition, we and others have had previous publications showing the presence of circulating epithelial cells of pancreatic origin from preneoplastic lesions (CITATIONS).

3.5. Survival analysis for LIN28B high PDAC CTCs has only been presented in univariate analysis. Has a multivariate analysis been calculated including confounding clinical parameters?

Thank you for the comment. The survival analyses were chosen after examination of the RNA sequencing data. Given the small size of the dataset, we did not perform multivariate analysis to identify confounding variables, but we fully intend to do so on subsequent larger studies in order to more rigorously evaluate our hypotheses.

3.6. Instead of analyzing three or more groups by one-way ANOVA followed by t-tests, the authors have to use one-way ANOVA with post-hoc Tukey's HSD test.

Thank you for the comment. The post-hoc Tukey test is indeed a valid method to mitigate the increased risk of type I error when evaluating multiple statistical tests. This essentially makes the acceptable p-value threshold more stringent in a manner that correlates with the square root of the number of comparisons. We forgot in the methods section to specify that the Bonferroni correction was used to correct p-values that were generated between groups after one-way ANOVA. The Bonferroni method mitigates the increase in type I error rate when multiple tests are done in a manner that correlates with the number of comparisons, which is more stringent than the post-Hoc Tukey test. We corrected this omission in the revised methods section.

3.7. Figure 5: single measurements in the boxplot for siGAPDH are missing.

Thank you for the observation. We have ensured that the final high-resolution figures are all transferred in a format that ensures no loss of image quality.

REFERENCES

1. Ligorio M, *et al.* Stromal Microenvironment Shapes the Intratumoral Architecture of Pancreatic Cancer. *Cell*, (2019).
2. Goldstein NS, Bassi D. Cytokeratins 7, 17, and 20 reactivity in pancreatic and ampulla of vater adenocarcinomas. Percentage of positivity and distribution is affected by the cut-point threshold. *Am J Clin Pathol* **115**, 695-702 (2001).
3. Jimenez RE, Z'Graggen K, Hartwig W, Graeme-Cook F, Warshaw AL, Fernandez-del Castillo C. Immunohistochemical characterization of pancreatic tumors induced by dimethylbenzanthracene in rats. *Am J Pathol* **154**, 1223-1229 (1999).
4. Kondratyeva LG, *et al.* Dependence of expression of regulatory master genes of embryonic development in pancreatic cancer cells on the intracellular concentration of the master regulator PDX1. *Dokl Biochem Biophys* **475**, 259-263 (2017).
5. Walsh N, *et al.* Identification of pancreatic cancer invasion-related proteins by proteomic analysis. *Proteome Sci* **7**, 3 (2009).
6. Zhang JS, Wang L, Huang H, Nelson M, Smith DI. Keratin 23 (K23), a novel acidic keratin, is highly induced by histone deacetylase inhibitors during differentiation of pancreatic cancer cells. *Genes Chromosomes Cancer* **30**, 123-135 (2001).

7. Ting DT, *et al.* Single-cell RNA sequencing identifies extracellular matrix gene expression by pancreatic circulating tumor cells. *Cell Rep* **8**, 1905-1918 (2014).
8. Marrinucci D, *et al.* Fluid biopsy in patients with metastatic prostate, pancreatic and breast cancers. *Phys Biol* **9**, 016003 (2012).
9. Schweizer J, *et al.* New consensus nomenclature for mammalian keratins. *J Cell Biol* **174**, 169-174 (2006).
10. Sierzega M, Mlynarski D, Tomaszewska R, Kulig J. Semiquantitative immunohistochemistry for mucin (MUC1, MUC2, MUC3, MUC4, MUC5AC, and MUC6) profiling of pancreatic ductal cell adenocarcinoma improves diagnostic and prognostic performance. *Histopathology* **69**, 582-591 (2016).
11. Jiang K, Tan E, Sayegh Z, Centeno B, Malafa M, Coppola D. Cancer Antigen 125 (CA125, MUC16) Protein Expression in the Diagnosis and Progression of Pancreatic Ductal Adenocarcinoma. *Appl Immunohistochem Mol Morphol* **25**, 620-623 (2017).
12. Hirono S, *et al.* Molecular markers associated with lymph node metastasis in pancreatic ductal adenocarcinoma by genome-wide expression profiling. *Cancer Sci* **101**, 259-266 (2010).
13. Kang MJ, Jang JY, Chang YR, Kwon W, Jung W, Kim SW. Revisiting the concept of lymph node metastases of pancreatic head cancer: number of metastatic lymph nodes and lymph node ratio according to N stage. *Ann Surg Oncol* **21**, 1545-1551 (2014).
14. Cancer Genome Atlas Research Network. Electronic address aadhe, Cancer Genome Atlas Research N. Integrated Genomic Characterization of Pancreatic Ductal Adenocarcinoma. *Cancer Cell* **32**, 185-203 e113 (2017).
15. Fachin F, *et al.* Monolithic Chip for High-throughput Blood Cell Depletion to Sort Rare Circulating Tumor Cells. *Sci Rep* **7**, 10936 (2017).
16. Franses JW, *et al.* Improved Detection of Circulating Epithelial Cells in Patients with Intraductal Papillary Mucinous Neoplasms. *Oncologist* **23**, 121-127 (2018).
17. Moffitt RA, *et al.* Virtual microdissection identifies distinct tumor- and stroma-specific subtypes of pancreatic ductal adenocarcinoma. *Nat Genet* **47**, 1168-1178 (2015).

REVIEWERS' COMMENTS:

Reviewer#1:

The authors have addressed my relatively minor concern and added substantial volume of data on other fronts as well.

Reviewer#2:

In response to the reviewer requests, the authors have added additional immunofluorescence data on LIN28B, and validation of the correlated gene sets in an additional cohort, among other updates. They have improved on the gene expression analyses with new supplementary figures and text. These changes have strengthened the conclusions of the paper. The majority of my questions have been addressed satisfactorily, but I list below my remaining concerns:

--The response to reviewers includes a detailed explanation of the choice of mucin genes as an indicator of bona fide CTC cells in their enriched population, including several additional references and analysis of these genes in the GTEX data. However, neither the references nor the GTEX analysis is included in the updated manuscript. I think it would improve the manuscript to include this information.

--Similarly, the analysis of the genes of interest in the previously published Ting et al. single cell paper would provide useful information to the reader, but is not included in the revision outside the reviewer response.

--I still find the conclusion and language used to call these correlated gene sets 'subtypes' problematic. The use of the word 'subtype' in gene expression analysis traditionally indicates that samples can be classified as one subtype or another. As shown in Figure 2B, this is not the case for the gene sets defined here. For example, locPDAC14 and locPDAC17 have high expression of both the LGALS3 and the WNT5A associated genes, and many samples, including metPDAC13 and metPDAC15 have high expression of both the LIN28B and KLF4 associated genes. The exploration of the stem-related genes in this data is definitely valuable, but the approach taken in this paper does not warrant calling these gene sets 'subtypes'. I suggest sticking to 'gene signatures', 'correlated gene sets' or any number of terms that actually reflect the analysis presented here.

--This conclusion in this sentence added on page 6, "CTC subpopulations or subtypes have different metastatic propensity that are linked with prognostic outcomes." is not supported by the data presented in this section. The lack of survival association from the mucin metesignature does not indicate that CTC subtypes are linked to prognostic outcomes. Please remove or clarify this conclusion.

--The heatmaps shown in Supp. Fig 3A are not well explained or all that convincing. The top heatmap track should be labeled. More information should be included to explain how the genes shown were selected. This explanatory sentence is confusing and should be reworded to improve clarification: "analysis of TCGA data of resected PDAC revealed correlation between LIN28B pathway markers HMGA2 and IGF2BP1/IGF2BP2/IGF2BP3 (Supplementary Fig. 3a) and poor clinical outcomes associated with high expression of let-7 target HMGA2 (Supplementary Fig. 3b), which should correlate with high levels of LIN28B."

--Please make sure that text in all figures, including supplementary ones, is clear and readable (i.e. S3c).

Reviewer#3:

The authors have addressed my comments and revised the manuscript.

June 1, 2020

Dear *Nature Communications* editorial staff,

We appreciate the additional comments from the reviewers and have responded in a point-by-point fashion to these critiques.

Best regards,

Joseph Franses, Julia Philipp, and David Ting

REVIEWER #1 COMMENTS:

The authors have addressed my relatively minor concern and added substantial volume of data on other fronts as well.

We appreciate deeply the positive assessment of our re-submitted work.

REVIEWER #2 COMMENTS:

In response to the reviewer requests, the authors have added additional immunofluorescence data on LIN28B, and validation of the correlated gene sets in an additional cohort, among other updates. They have improved on the gene expression analyses with new supplementary figures and text. These changes have strengthened the conclusions of the paper. The majority of my questions have been addressed satisfactorily, but I list below my remaining concerns:

--The response to reviewers includes a detailed explanation of the choice of mucin genes as an indicator of bona fide CTC cells in their enriched population, including several additional references and analysis of these genes in the GTEX data. However, neither the references nor the GTEX analysis is included in the updated manuscript. I think it would improve the manuscript to include this information.

We appreciate this feedback and have:

1. Added the additional references regarding MUC gene expression in PDAC in the main manuscript. They are now listed as references 19-22 in the revised document.
2. Added our GTEX analysis as a new Supplementary Fig. 1b. We cited the details of accessing the GTEX portal in the supplementary methods section (page 9 of supplement).

--Similarly, the analysis of the genes of interest in the previously published Ting et al. single cell paper would provide useful information to the reader, but is not included in the revision outside the reviewer response.

We appreciate this feedback and have added the heatmap utilizing single-cell data from the 2014 publication (Ting et al, Cell Rep 2014) as a new Supplementary Fig. 5b. We refer to this sub-figure in the revised manuscript (page 9, paragraph 1).

--I still find the conclusion and language used to call these correlated gene sets 'subtypes' problematic. The use of the word 'subtype' in gene expression analysis traditionally indicates that samples can be classified as one subtype or another. As shown in Figure 2B, this is not the case for the gene sets defined here. For example, locPDAC14 and locPDAC17 have high expression of both the LGALS3 and the WNT5A associated genes, and many samples, including metPDAC13 and metPDAC15 have high expression of both the LIN28B and KLF4 associated genes. The exploration of the stem-related genes in this data is definitely valuable, but the approach taken in this paper does not warrant calling these gene sets 'subtypes'. I suggest sticking to 'gene signatures', 'correlated gene sets' or any number of terms that actually reflect the analysis presented here.

We appreciate this feedback and have amended the language in the revised manuscript to read as “correlated gene set” rather than subtype.

--This conclusion in this sentence added on page 6, “CTC subpopulations or subtypes have different metastatic propensity that are linked with prognostic outcomes.” is not supported by the data presented in this section. The lack of survival association from the mucin metaskinature does not indicate that CTC subtypes are linked to prognostic outcomes. Please remove or clarify this conclusion.

We appreciate this feedback and have deleted the quoted part of the sentence. The wording on the revised manuscript document now reads: “The mucin metaskinatures did not predict survival in our cohort of patients (Supplementary Fig. 1d), suggesting that the presence of CTCs is not directly linked with clinical outcome.”

--The heatmaps shown in Supp. Fig 3A are not well explained or all that convincing. The top heatmap track should be labeled. More information should be included to explain how the genes shown were selected. This explanatory sentence is confusing and should be reworded to improve clarification: “analysis of TCGA data of resected PDAC revealed correlation between LIN28B pathway markers HMGA2 and IGF2BP1/IGF2BP2/IGF2PB3 (Supplementary Fig. 3a) and poor clinical outcomes associated with high expression of let-7 target HMGA2 (Supplementary Fig. 3b), which should correlate with high levels of LIN28B.”

We appreciate this feedback. The point we attempting to make with Supplementary Fig. 3a is that genes that should be positively regulated by LIN28B are indeed correlated in the TCGA PDAC dataset. In order to perform a simple analysis, we utilized Z-score normalized TCGA data using the Cbioportal “Next-Generation Clustered Heat Map” function (Bradley et al, Cancer Research 2017). The point of Supplementary Fig. 3b is to show association of high *HMGA2* expression in TCGA PDAC tissue with poor clinical outcome.

We have therefore implemented the following changes:

1. We have cited the Cbioportal resource in the supplement.
2. For the Supplementary Fig. 3a legend, we included more details about how the heat map was generated.
3. The previously-confusing statements now read (on page 8, paragraph 1 in the new manuscript): “Analysis of TCGA data from resected PDAC revealed correlated expression of LIN28B pathway genes HMGA2 and IGF2BP1/2/3 (Supplementary Fig. 3a), supporting the notion of coordinate expression of these genes driven by LIN28B. Further, high expression of HMGA2 in this dataset correlated with poor clinical outcomes (Supplementary Fig. 3b). However, the correlated gene families that we identified in our CTC data were not observed in the TCGA dataset (Supplementary Fig. 3c), which likely stems from differences between primary tumour and CTC cellular heterogeneity as we had demonstrated previously with single cell RNA-seq.”

--Please make sure that text in all figures, including supplementary ones, is clear and readable (i.e. S3c).

We appreciate this feedback. We have improved the clarity of Supplementary Fig. 3c and have checked to ensure that all other figures are of publication quality

REVIEWER #3 COMMENTS:

The authors have addressed my comments and revised the manuscript.

We appreciate deeply the positive assessment of our re-submitted work.